# Molecule Generation with Fragment Retrieval Augmentation

**Seul Lee**[1]* **Karsten Kreis**[2] **Srimukh Prasad Veccham**[2] **Meng Liu**[2]
**Danny Reidenbach**[2] **Saee Paliwal**[2] **Arash Vahdat**[2]† **Weili Nie**[2]†
[1]KAIST [2]NVIDIA
seul.lee@kaist.ac.kr
{kkreis,sveccham,menliu,dreidenbach,saeep,avahdat,wnie}@nvidia.com

## Abstract

Fragment-based drug discovery, in which molecular fragments are assembled into new molecules with desirable biochemical properties, has achieved great success. However, many fragment-based molecule generation methods show limited exploration beyond the existing fragments in the database as they only reassemble or slightly modify the given ones. To tackle this problem, we propose a new fragment-based molecule generation framework with retrieval augmentation, namely *Fragment Retrieval-Augmented Generation* ($f$-RAG). $f$-RAG is based on a pre-trained molecular generative model that proposes additional fragments from input fragments to complete and generate a new molecule. Given a fragment vocabulary, $f$-RAG retrieves two types of fragments: (1) *hard fragments*, which serve as building blocks that will be explicitly included in the newly generated molecule, and (2) *soft fragments*, which serve as reference to guide the generation of new fragments through a trainable *fragment injection module*. To extrapolate beyond the existing fragments, $f$-RAG updates the fragment vocabulary with generated fragments via an iterative refinement process which is further enhanced with post-hoc genetic fragment modification. $f$-RAG can achieve an improved exploration-exploitation trade-off by maintaining a pool of fragments and expanding it with novel and high-quality fragments through a strong generative prior.

## 1 Introduction

The goal of small molecule drug discovery is to discover molecules with specific biochemical target properties, such as synthesizability [9], non-toxicity [40], solubility [31] and binding affinity, in the vast chemical space. Fragment-based drug discovery (FBDD) has been considered as an effective approach to explore the chemical space and has resulted in many successful marketed drugs [27]. Contrary to high-throughput screening (HTS) [41] that searches from a library of drug-like molecules, FBDD constructs a library of molecular fragments to synthesize new molecules beyond the existing molecule library, leading to a better chemical coverage [5].

Recently, generative models have been adopted in the field of FBDD to accelerate the process of searching for drug

Figure 1: **A radar plot of target properties.** $f$-RAG strikes better balance among optimization performance, diversity, novelty, and synthesizability than the state-of-the-art techniques on the PMO benchmark [10].

---

*Work done during an internship at NVIDIA.
†Equal advising.

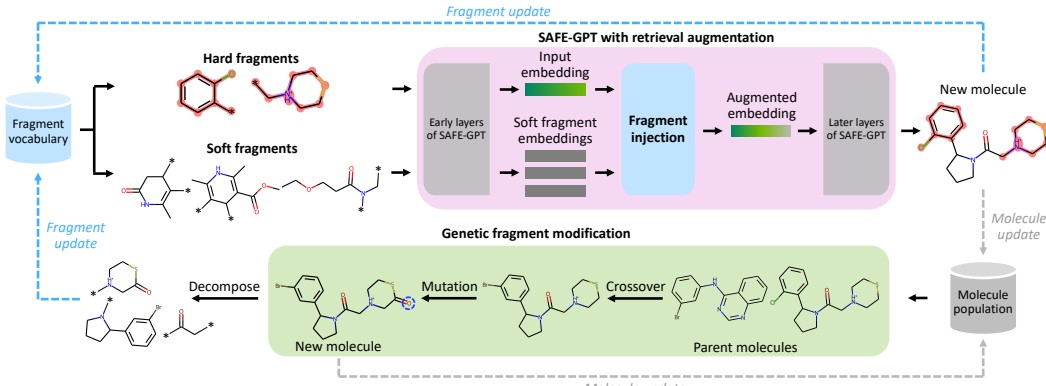

Figure 2: **The overall framework of $f$-RAG.** After an initial fragment vocabulary is constructed from an existing molecule library, two types of fragments are retrieved during generation. Hard fragments are explicitly included in the newly generated molecules, while soft fragments implicitly guide the generation of new fragments. SAFE-GPT generates a molecule using hard fragments as input, while the fragment injection module in the middle of the SAFE-GPT layers injects the embeddings of soft fragments into the input embedding. After the generation, the molecule population and fragment vocabulary are updated with the newly generated molecule and its fragments, respectively. The exploration is further enhanced with genetic fragment modification, which also updates the fragment vocabulary and molecule population.

candidates [18, 45, 46, 30, 20, 11, 25]. These methods reassemble or slightly modify the fragments to generate new molecules with the knowledge of the generative model. As the methods are allowed to exploit chemical knowledge in the form of fragments to narrow the search space, they have shown meaningful success in generating optimized molecules. However, their performance is largely limited by the existing library of molecular fragments, where discovering new fragments is either impossible or highly limited. Some of the methods [14, 38, 25] suggested making modifications to existing fragments, but the modifications are only applied to small and local substructures and still heavily dependent on the existing fragments. The limited exploration beyond known fragments greatly hinders the possibility of generating diverse and novel drug candidates that may exhibit better target properties. Therefore, it is a challenge to improve the ability of discovering novel high-quality fragments, while exploiting existing chemical space effectively.

To this end, we propose a fragment-based molecule generation framework leveraging retrieval-augmented generation (RAG) [26], namely *Fragment Retrieval-Augmented Generation* ($f$-RAG). As shown in Figure 2, $f$-RAG augments the pre-trained molecular language model SAFE-GPT [34] with two types of retrieved fragments: *hard fragments* and *soft fragments* for a better exploration-exploitation trade-off. Specifically, we first construct a fragment vocabulary by decomposing known molecules from the existing library into fragments and scoring the fragments by measuring their contribution to target properties. From the fragment vocabulary, $f$-RAG first retrieves fragments that will be explicitly included in the new molecule (i.e., hard fragments). Hard fragments serve as the input context to the molecular language model that predicts the remaining fragments. In addition to retrieval of hard fragments, $f$-RAG retrieves fragments that will not be part of the generated molecule but provide informative guidance on predicting novel, diverse molecules (i.e., soft fragments). Concretely, the embeddings of soft fragments are fused with the embeddings of hard fragments (or input embeddings) through a lightweight *fragment injection module* in the middle of SAFE-GPT. The fragment injection module allows SAFE-GPT to generate new fragments by referring to the information conveyed by soft fragments.

During training, we only update the fragment injection module while keeping SAFE-GPT frozen. Inspired by Wang et al. [42], we train $f$-RAG to learn how to leverage the retrieved fragments for molecule generation, using a self-supervised loss that predicts the most similar one in the set of soft fragments. At inference, $f$-RAG dynamically updates the fragment vocabulary with newly generated fragments via an iterative refinement process. To further enhance exploration, we propose to modify the generated fragments from SAFE-GPT with a post-hoc genetic fragment modification process. The proposed $f$-RAG takes the advantages of both hard and soft fragment retrieval to achieve an improved exploration-exploitation trade-off. We verify $f$-RAG on various molecular optimization tasks, by examining the optimization performance, along with diversity, novelty, and

synthesizability. As shown in Figure 1, $f$-RAG exhibits the best balance across these essential considerations, demonstrating its applicability as a promising tool for drug discovery.

We summarize our contributions as follows:

- We introduce $f$-RAG, a novel molecular generative framework that combines FBDD and RAG.
- We propose a retrieval augmentation strategy that operates at the fragment level with two types of fragments, allowing fine-grained guidance to achieve an improved exploration-exploitation trade-off and generate high-quality drug candidates.
- Through extensive experiments, we demonstrate the effectiveness of $f$-RAG in various drug discovery tasks that simulate real-world scenarios.

## 2 Related Work

**Fragment-based molecule generation.** Fragment-based molecular generative models refer to a class of methods that reassemble existing molecular substructures (i.e., fragments) to generate new molecules. Jin et al. [18] proposed to find big molecular substructures that satisfy the given chemical properties, and learn to complete the substructures into final molecules by adding small branches. Xie et al. [45] proposed to progressively add or delete fragments using Markov chain Monte Carlo (MCMC) sampling. Yang et al. [46] proposed to use reinforcement learning (RL) to assemble fragments, while Maziarz et al. [30], Kong et al. [20], and Geng et al. [11] used VAE-based techniques. Lee et al. [25] proposed to take the target chemical properties into account in the fragment vocabulary construction and used a combination of RL and a genetic algorithm (GA) to assemble and modify the fragments. On the other hand, graph-based GAs [14, 38] decompose parent molecules into fragments that are combined to generate an offspring molecule, and are also mutated with a small probability. Since fragment-based strategies are limited by generating molecules outside of the possible combinations of existing fragments, they suffer from limited exploration in the chemical space. Some of the methods [14, 38, 25] suggest making modifications to existing fragments to overcome this problem, but this is not a fundamental solution because the modifications are only local, still being based on the existing fragments.

**Retrieval-augmented molecule generation.** Retrieval-augmented generation (RAG) [26] refers to a technique that retrieves context from external data databases to guide the generation of a generative model. Recently, RAG has gained attention as a means to enhance accuracy and reliability of large language models (LLMs) [6, 36, 3]. In the field of molecule optimization, Liu et al. [28] developed a text-based drug editing framework that repurposes a conversational language model for solving molecular tasks, where the domain knowledge is injected by a retrieval and feedback module. More related to us, Wang et al. [42] proposed to use a pre-trained molecular language model to generate molecules, while augmenting the generation with retrieved molecules that have high target properties. Contrary to this method that retrieves molecules, our proposed $f$-RAG employs a fine-grained retrieval augmentation scheme that operates at the fragment level to achieve an improved exploration-exploitation trade-off.

## 3 Method

In this section, we introduce our Fragment Retrieval-Augmented Generation ($f$-RAG) framework. $f$-RAG aims to generate optimized molecules by leveraging existing chemical knowledge through RAG, while exploring beyond the known chemical space under the fragment-based drug discovery (FBDD) paradigm. We first introduce the hard fragment retrieval in Sec. 3.1. Then, we present the soft fragment retrieval in Sec. 3.2. Lastly, we describe the genetic fragment modification in Sec. 3.3.

### 3.1 Hard Fragment Retrieval

Given a set of $N$ molecules $x_i$ and their corresponding properties $y_i \in [0, 1]$, denoted as $\mathcal{D} = \{(x_i, y_i)\}_{i=1}^N$, we first construct a fragment vocabulary. We adopt an arm-linker-arm slicing algorithm provided by Noutahi et al. [34] which decomposes a molecule $x$ into three fragments: two arms $F_{\text{arm}}$ (i.e., fragments that have one attachment point) and one linker $F_{\text{linker}}$ (i.e., a fragment that has two attachment points). Decomposing molecules into arms and linkers (or scaffolds) is a popular approach

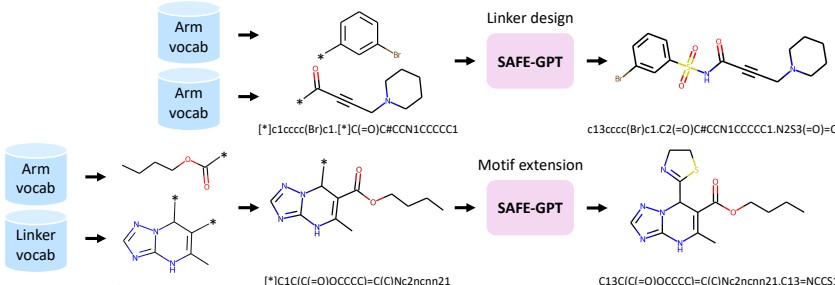

Figure 3: **Hard fragment retrieval** of $f$-RAG. With a probability of $50\%$, $f$-RAG either retrieves two arms as hard fragments for linker design (**top**) or one arm and one linker as hard fragments for motif extension (**bottom**).

utilized in various drug discovery strategies, such as scaffold decoration or scaffold hopping [37]. We ignored the molecules in the training set that cannot be decomposed, and a set of arms $\mathcal{F}_{\text{arm}} = \{F_{\text{arm},j}\}_{j=1}^{2N}$ and a set of linkers $\mathcal{F}_{\text{linker}} = \{F_{\text{linker},j}\}_{j=1}^{N}$ are obtained after the algorithm is applied to the molecules $\{x_i\}_{i=1}^{N}$. Subsequently, we calculate the score of each fragment $F_j \in \mathcal{F}_{\text{arm}} \cup \mathcal{F}_{\text{linker}}$ using the average property of all molecules containing $F_j$ as their substructure as follows:

$$\text{score}(F_j) = \frac{1}{|S(F_j)|} \sum_{(x,y) \in S(F_j)} y, \tag{1}$$

where $\text{score}(F_j) \in [0,1]$, and $S(F_j) = \{(x,y) \in \mathcal{D} : F_j \text{ is a fragment of } x\}$. Intuitively, the fragment score evaluates the contribution of a given fragment to the target property of the whole molecule of which it is a part. From $\mathcal{F}_{\text{arm}}$ and $\mathcal{F}_{\text{linker}}$, we choose the top-$N_{\text{frag}}$ fragments based on the score to construct the arm fragment vocabulary $\mathcal{V}_{\text{arm}} \subset \mathcal{F}_{\text{arm}}$ and the linker fragment vocabulary $\mathcal{V}_{\text{linker}} \subset \mathcal{F}_{\text{linker}}$, respectively.

Given the fragment vocabularies $\mathcal{V}_{\text{arm}}$ and $\mathcal{V}_{\text{linker}}$ consisting of high-property fragments, two hard fragments are randomly retrieved from the vocabularies. The hard fragments together form a partial molecular sequence that serves as input to a pre-trained molecular language model. In this work, we employ Sequential Attachment-based Fragment Embedding (SAFE) [34] as the molecular representation and SAFE-GPT [34] as the backbone generative model of $f$-RAG. SAFE is a non-canonical version of simplified molecular-input line-entry system (SMILES) [43] that represents molecules as a sequence of dot-connected fragments. Importantly, the order of fragments in a SAFE string does not affect the molecular identity. Using the SAFE representation, $f$-RAG forces the hard fragments to be included in a newly generated molecule by providing them as an input sequence to the language model to complete the rest of the sequence.

During generation, with a probability of $50\%$, $f$-RAG either (1) retrieves two hard fragments from $\mathcal{V}_{\text{arm}}$ or (2) retrieves one from $\mathcal{V}_{\text{arm}}$ and one from $\mathcal{V}_{\text{linker}}$. In the former case, $f$-RAG performs linker design, which generates a new fragment that links the input fragments. In the latter case, $f$-RAG first randomly selects an attachment point in the retrieved linker and combines it with the retrieved arm to form a single fragment, and then performs motif extension, which generates a new fragment that completes the molecule (Figure 3).

### 3.2 Soft Fragment Retrieval

Given two hard fragments as input, the molecular language model generates one new fragment to complete a molecule. Instead of relying solely on the model to generate the new fragment, we propose to augment the generation with the information of $K$ retrieved fragments, which we refer to as soft fragments, to guide the generation. Specifically, if the two hard fragments are all arms, $f$-RAG randomly retrieves soft fragments from $\mathcal{V}_{\text{linker}}$. If one of the hard fragments is an arm and another is a linker, $f$-RAG randomly retrieves soft fragments from $\mathcal{V}_{\text{arm}}$. Using up to the $L$-th layer of the language model $\text{LM}^{0:L}$, the embeddings of the input sequence $x_{\text{input}}$ and the soft fragments $\{F_{\text{soft},k}\}_{k=1}^{K}$ are obtained as follows:

$$\boldsymbol{h}_{\text{input}} = \text{LM}^{0:L}(x_{\text{input}}) \text{ and } \boldsymbol{H}_{\text{soft}} = \text{concatenate}([\boldsymbol{h}_{\text{soft}}^1, \boldsymbol{h}_{\text{soft}}^2, \ldots, \boldsymbol{h}_{\text{soft}}^K]),$$
$$\text{where } \boldsymbol{h}_{\text{soft}}^k = \text{LM}^{0:L}(F_{\text{soft},k}) \text{ for } k=1,2,\ldots,K. \tag{2}$$

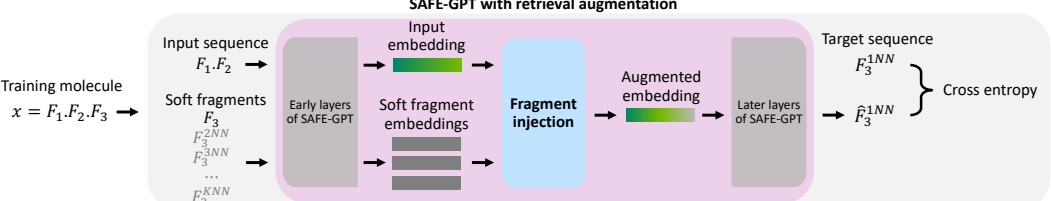

Figure 4: **The self-supervised training process of the fragment injection module of $f$-RAG.** $F^{k\text{NN}}$ denotes the $k$-th most similar fragment to $F$. Using $F_1$ and $F_2$ as hard fragments, while using $F_3$ and its neighbors $\{F_3^{kNN}\}_{k=2}^K$ as soft fragments, the training objective is to predict $F_3^{1NN}$.

Subsequently, $f$-RAG injects the embeddings of soft fragments through a trainable *fragment injection module*. Following Wang et al. [42], the fragment injection module uses cross-attention to fuse the embeddings of the input sequence and soft fragments as follows:

$$\boldsymbol{h} = \text{FI}(\boldsymbol{h}_{\text{input}}, \boldsymbol{H}_{\text{soft}}) = \text{softmax}\left(\frac{\text{Query}(\boldsymbol{h}_{\text{input}}) \cdot \text{Key}(\boldsymbol{H}_{\text{soft}})^\top}{\sqrt{d_{\text{Key}}}}\right) \cdot \text{Value}(\boldsymbol{H}_{\text{soft}}), \qquad (3)$$

where FI is the fragment injection module, Query, Key, and Value are multi-layer perceptrons (MLPs), and $d_{\text{Key}}$ is the output dimension of Key. Next, a molecule is generated by decoding the augmented embedding $\boldsymbol{h}$ using the later layers of the language model as $x_{\text{new}} = \text{LM}^{L+1:L_T}(\boldsymbol{h})$, where $L_T$ is the total number of layers of the model. With the fragment injection module, $f$-RAG can utilize information of soft fragments to generate novel fragments which are also likely to contribute to the high target properties. During generation, the fragment vocabulary is dynamically updated through an iterative process that scores newly generated fragments based on Eq. (1) and replaces fragments in the fragment vocabulary to the top-$N_{\text{frag}}$ fragments.

Next, we need to train $f$-RAG to learn how to augment the retrieved soft fragments into the molecule generation. To retain the high generation quality of SAFE-GPT and make the training process efficient, we keep the backbone language model frozen and only train the lightweight fragment injection module. Inspired by Wang et al. [42], we propose a new self-supervised objective that predicts the most similar fragment to the input fragments. Specifically, each molecular sequence $x$ in the training set is first decomposed into fragment sequences $(F_1, F_2, F_3)$ with a random permutation between the fragments, using the same slicing algorithm used in the vocabulary construction. Importantly, using the SAFE representation, $x$ can be simply represented by connecting its fragments with dots as $F_1.F_2.F_3$. We consider the first two fragments as hard fragments. Given the remaining fragment $F_3$, we retrieve its $K$ most similar fragments $\{F_3^{kNN}\}_{k=1}^K$ from the training fragment pool. Here, we use the pairwise Tanimoto similarity using Morgan fingerprints of radius 2 and 1024 bits. Using the hard fragments as the input sequence as $F_1.F_2$, the objective is to predict the most similar fragment $F_3^{1NN}$ utilizing the original fragment and the next $K-1$ most similar fragments $\{F_3^{kNN}\}_{k=2}^K$ as the soft fragments. The training process is illustrated in Figure 4, and the details are provided in Section D.1.

Note that the training of the fragment injection module is target property-agnostic, as the fragments used for training are independent of the target property. In contrast, the fragment vocabularies used for generation are target property-specific, as it is constructed using the scoring function in Eq. (1). This allows $f$-RAG to effectively generate optimized molecules across different target properties without any retraining.

### 3.3 Genetic Fragment Modification

To further enhance exploration in the chemical space, we propose to modify the generated fragments with a post-hoc genetic algorithm (GA). Specifically, we adopt the operations of Graph GA [14]. We first initialize the population $\mathcal{P}$ with the top-$N_{\text{mol}}$ molecules generated by our fragment retrieval-augmented SAFE-GPT based on the target property $y$. Parent molecules are then randomly selected from the population and offspring molecules are generated by the crossover and mutation operations (see Jensen [14] for more details). The offspring molecules can have new fragments not contained in the initial fragment vocabulary, and the fragment vocabularies $\mathcal{V}_{\text{arm}}$ and $\mathcal{V}_{\text{linker}}$ are again updated by the top-$N_{\text{frag}}$ fragments based on the scores of Eq. (1). In the subsequent generation, the population $\mathcal{P}$ is updated with the generated molecules so far by both SAFE-GPT and GA.

As shown in Figure 2, $f$-RAG generates desirible molecules through multiple cycles of (1) the SAFE-GPT generation augmented with the hard fragment retrieval (Section 3.1) and the soft fragment retrieval (Section 3.2), and (2) the GA generation (Section 3.3). Through this interplay of hard fragment retrieval, soft fragment retrieval, and the genetic fragment modification, $f$-RAG can exploit existing chemical knowledge through the form of fragments both explicitly and implicitly, while exploring beyond initial fragments by the dynamic vocabulary update. We summarize the generation process of $f$-RAG in Algorithm 1 in Section C.

## 4 Experiments

We validate $f$-RAG on molecule generation tasks that simulate various real-world drug discovery problems. We first conduct experiments on the practical molecular optimization (PMO) benchmark [10] in Section 4.1. We then conduct experiments to generate novel molecules that have high binding affinity, drug-likeness, and synthesizability in Section 4.2. We further perform analyses in Section 4.3.

### 4.1 Experiments on PMO Benchmark

**Setup.** We demonstrate the efficacy of $f$-RAG on the 23 tasks from the PMO benchmark. Following the standard setting of the benchmark, we set the maximum number of oracle calls to 10,000 and evaluate **optimization performance** with the area under the curve (AUC) of the average property scores of the top-10 molecules versus oracle calls. In addition, we evaluate **diversity**, **novelty**, and **synthesizability** of generated molecules, other essential considerations in drug discovery. We use the Therapeutics Data Commons (TDC) library [12] to calculate diversity. Following previous works [18, 45, 24, 25], novelty is defined as the fraction of molecules that have the maximum Tanimoto similarity less than $0.4$ with the training molecules. We use the synthetic accessibility (SA) [8] score of the TDC library to measure synthesizability. These values are measured using the top-100 generated molecules, following the setting of the benchmark. Further explanation on the evaluation metrics and experimental setup are provided in Section D.3.

**Baselines.** We employ the top-7 methods reported by the PMO benchmark and two recent state-of-the-art methods as our baselines. **Graph GA** [14] is a GA with a fragment-level crossover operation, and **Mol GA** [38] is a more hyperparameter-tuned version of Graph GA. **Genetic GFN** [19] is a method that uses Graph GA to guide generation of a GFlowNet. **REINVENT** [35] is a SMILES-based RL model and **SELFIES-REINVENT** is a modified REINVENT that uses the self-referencing embedded strings (SELFIES) [21] representation. **GP BO** [39] is a method that optimizes the Gaussian process acquisition function with Graph GA. **STONED** [33] is a GA-based model that operates on SELFIES strings. **LSTM HC** [7] is a SMILES-based LSTM model. **SMILES GA** [47] is a GA whose genetic operations are based on the SMILES context-free grammar.

**Results.** The results of optimization performance are shown in Table 1. $f$-RAG outperforms the previous methods in terms of the sum of the AUC top-10 values and achieves the highest AUC top-10 values in 12 out of 23 tasks, demonstrating that the proposed combination of hard fragment retrieval, soft fragment retrieval, and genetic fragment modification is highly effective in discovering optimized drug candidates that have high target properties. On the other hand, the average scores of diversity, novelty, and synthesizability of the generated molecules are summarized in Table 2, and the full results are presented in Tables 4, 5, and 6. As shown in these tables, $f$-RAG achieves the best diversity and synthesizability, and the second best novelty. Notably, $f$-RAG shows the highest diversity in 12 out of 23 tasks, and the highest synthesizability in 19 out of 23 tasks. Note that the high novelty value of Mol GA comes at the cost of other important factors, i.e., optimization performance, diversity, and synthesizability. The essential considerations in drug discovery often conflict with each other, making the drug discovery problem challenging. These trade-offs are also visualized in Figure 1 and Figure 7, and $f$-RAG effectively improves the trade-offs by utilizing existing fragments while dynamically updating the fragment vocabulary with newly proposed fragments.

### 4.2 Optimization of Docking Score under QED, SA, and Novelty Constraints

**Setup.** Following Lee et al. [24] and Lee et al. [25], we validate $f$-RAG in a set of tasks that aim to optimize the binding affinity against a target protein while also maintaining high drug-likeness,

Table 1: **PMO AUC top-10 results.** The results are the mean and standard deviation of 3 independent runs. The results for Genetic GFN [19] and Mol GA [38] are taken from the respective original papers and the results for other baselines are taken from Gao et al. [10]. The best results are highlighted in bold.

| Oracle | $f$-RAG (ours) | Genetic GFN | Mol GA | REINVENT | Graph GA |
|---|---|---|---|---|---|
| albuterol_similarity | **0.977** $\pm$ 0.002 | 0.949 $\pm$ 0.010 | 0.896 $\pm$ 0.035 | 0.882 $\pm$ 0.006 | 0.838 $\pm$ 0.016 |
| amlodipine_mpo | 0.749 $\pm$ 0.019 | **0.761** $\pm$ 0.019 | 0.688 $\pm$ 0.039 | 0.635 $\pm$ 0.035 | 0.661 $\pm$ 0.020 |
| celecoxib_rediscovery | 0.778 $\pm$ 0.007 | **0.802** $\pm$ 0.029 | 0.567 $\pm$ 0.083 | 0.713 $\pm$ 0.067 | 0.630 $\pm$ 0.097 |
| deco_hop | **0.936** $\pm$ 0.011 | 0.733 $\pm$ 0.109 | 0.649 $\pm$ 0.025 | 0.666 $\pm$ 0.044 | 0.619 $\pm$ 0.004 |
| drd2 | **0.992** $\pm$ 0.000 | 0.974 $\pm$ 0.006 | 0.936 $\pm$ 0.016 | 0.945 $\pm$ 0.007 | 0.964 $\pm$ 0.012 |
| fexofenadine_mpo | **0.856** $\pm$ 0.016 | **0.856** $\pm$ 0.039 | 0.825 $\pm$ 0.019 | 0.784 $\pm$ 0.006 | 0.760 $\pm$ 0.011 |
| gsk3b | **0.969** $\pm$ 0.003 | 0.881 $\pm$ 0.042 | 0.843 $\pm$ 0.039 | 0.865 $\pm$ 0.043 | 0.788 $\pm$ 0.070 |
| isomers_c7h8n2o2 | 0.955 $\pm$ 0.008 | **0.969** $\pm$ 0.003 | 0.878 $\pm$ 0.026 | 0.852 $\pm$ 0.036 | 0.862 $\pm$ 0.065 |
| isomers_c9h10n2o2pf2cl | 0.850 $\pm$ 0.005 | **0.897** $\pm$ 0.007 | 0.865 $\pm$ 0.012 | 0.642 $\pm$ 0.054 | 0.719 $\pm$ 0.047 |
| jnk3 | **0.904** $\pm$ 0.004 | 0.764 $\pm$ 0.069 | 0.702 $\pm$ 0.123 | 0.783 $\pm$ 0.023 | 0.553 $\pm$ 0.136 |
| median1 | 0.340 $\pm$ 0.007 | **0.379** $\pm$ 0.010 | 0.257 $\pm$ 0.009 | 0.356 $\pm$ 0.009 | 0.294 $\pm$ 0.021 |
| median2 | **0.323** $\pm$ 0.005 | 0.294 $\pm$ 0.007 | 0.301 $\pm$ 0.021 | 0.276 $\pm$ 0.008 | 0.273 $\pm$ 0.009 |
| mestranol_similarity | 0.671 $\pm$ 0.021 | **0.708** $\pm$ 0.057 | 0.591 $\pm$ 0.053 | 0.618 $\pm$ 0.048 | 0.579 $\pm$ 0.022 |
| osimertinib_mpo | **0.866** $\pm$ 0.009 | 0.860 $\pm$ 0.008 | 0.844 $\pm$ 0.015 | 0.837 $\pm$ 0.009 | 0.831 $\pm$ 0.005 |
| perindopril_mpo | **0.681** $\pm$ 0.017 | 0.595 $\pm$ 0.014 | 0.547 $\pm$ 0.022 | 0.537 $\pm$ 0.016 | 0.538 $\pm$ 0.009 |
| qed | 0.939 $\pm$ 0.001 | **0.942** $\pm$ 0.000 | 0.941 $\pm$ 0.001 | 0.941 $\pm$ 0.000 | 0.940 $\pm$ 0.000 |
| ranolazine_mpo | **0.820** $\pm$ 0.016 | 0.819 $\pm$ 0.018 | 0.804 $\pm$ 0.011 | 0.760 $\pm$ 0.009 | 0.728 $\pm$ 0.012 |
| scaffold_hop | 0.576 $\pm$ 0.014 | **0.615** $\pm$ 0.100 | 0.527 $\pm$ 0.025 | 0.560 $\pm$ 0.019 | 0.517 $\pm$ 0.007 |
| sitagliptin_mpo | 0.601 $\pm$ 0.011 | **0.634** $\pm$ 0.039 | 0.582 $\pm$ 0.040 | 0.021 $\pm$ 0.003 | 0.433 $\pm$ 0.075 |
| thiothixene_rediscovery | **0.584** $\pm$ 0.009 | 0.583 $\pm$ 0.034 | 0.519 $\pm$ 0.041 | 0.534 $\pm$ 0.013 | 0.479 $\pm$ 0.025 |
| troglitazone_rediscovery | 0.448 $\pm$ 0.017 | **0.511** $\pm$ 0.054 | 0.427 $\pm$ 0.031 | 0.441 $\pm$ 0.032 | 0.390 $\pm$ 0.016 |
| valsartan_smarts | **0.627** $\pm$ 0.058 | 0.135 $\pm$ 0.271 | 0.000 $\pm$ 0.000 | 0.178 $\pm$ 0.358 | 0.000 $\pm$ 0.000 |
| zaleplon_mpo | 0.486 $\pm$ 0.004 | **0.552** $\pm$ 0.033 | 0.519 $\pm$ 0.029 | 0.358 $\pm$ 0.062 | 0.346 $\pm$ 0.032 |
| Sum | **16.928** | 16.213 | 14.708 | 14.196 | 13.751 |

| Oracle | SELFIES-REINVENT | GP BO | STONED | LSTM HC | SMILES GA |
|---|---|---|---|---|---|
| albuterol_similarity | 0.826 $\pm$ 0.030 | 0.898 $\pm$ 0.014 | 0.745 $\pm$ 0.076 | 0.719 $\pm$ 0.018 | 0.661 $\pm$ 0.066 |
| amlodipine_mpo | 0.607 $\pm$ 0.014 | 0.583 $\pm$ 0.044 | 0.608 $\pm$ 0.046 | 0.593 $\pm$ 0.016 | 0.549 $\pm$ 0.009 |
| celecoxib_rediscovery | 0.573 $\pm$ 0.043 | 0.723 $\pm$ 0.053 | 0.382 $\pm$ 0.041 | 0.539 $\pm$ 0.018 | 0.344 $\pm$ 0.027 |
| deco_hop | 0.631 $\pm$ 0.012 | 0.629 $\pm$ 0.018 | 0.611 $\pm$ 0.008 | 0.826 $\pm$ 0.017 | 0.611 $\pm$ 0.006 |
| drd2 | 0.943 $\pm$ 0.005 | 0.923 $\pm$ 0.017 | 0.913 $\pm$ 0.020 | 0.919 $\pm$ 0.015 | 0.908 $\pm$ 0.019 |
| fexofenadine_mpo | 0.741 $\pm$ 0.002 | 0.722 $\pm$ 0.005 | 0.797 $\pm$ 0.016 | 0.725 $\pm$ 0.003 | 0.721 $\pm$ 0.015 |
| gsk3b | 0.780 $\pm$ 0.037 | 0.851 $\pm$ 0.041 | 0.668 $\pm$ 0.049 | 0.839 $\pm$ 0.015 | 0.629 $\pm$ 0.044 |
| isomers_c7h8n2o2 | 0.849 $\pm$ 0.034 | 0.680 $\pm$ 0.117 | 0.899 $\pm$ 0.011 | 0.485 $\pm$ 0.045 | 0.913 $\pm$ 0.021 |
| isomers_c9h10n2o2pf2cl | 0.733 $\pm$ 0.029 | 0.469 $\pm$ 0.180 | 0.805 $\pm$ 0.031 | 0.342 $\pm$ 0.027 | 0.860 $\pm$ 0.065 |
| jnk3 | 0.631 $\pm$ 0.064 | 0.564 $\pm$ 0.155 | 0.523 $\pm$ 0.092 | 0.661 $\pm$ 0.039 | 0.316 $\pm$ 0.022 |
| median1 | 0.355 $\pm$ 0.011 | 0.301 $\pm$ 0.014 | 0.266 $\pm$ 0.016 | 0.255 $\pm$ 0.010 | 0.192 $\pm$ 0.012 |
| median2 | 0.255 $\pm$ 0.005 | 0.297 $\pm$ 0.009 | 0.245 $\pm$ 0.032 | 0.248 $\pm$ 0.008 | 0.198 $\pm$ 0.005 |
| mestranol_similarity | 0.620 $\pm$ 0.029 | 0.627 $\pm$ 0.089 | 0.609 $\pm$ 0.101 | 0.526 $\pm$ 0.032 | 0.469 $\pm$ 0.029 |
| osimertinib_mpo | 0.820 $\pm$ 0.003 | 0.787 $\pm$ 0.006 | 0.822 $\pm$ 0.012 | 0.796 $\pm$ 0.002 | 0.817 $\pm$ 0.011 |
| perindopril_mpo | 0.517 $\pm$ 0.021 | 0.493 $\pm$ 0.011 | 0.488 $\pm$ 0.011 | 0.489 $\pm$ 0.007 | 0.447 $\pm$ 0.013 |
| qed | 0.940 $\pm$ 0.000 | 0.937 $\pm$ 0.000 | 0.941 $\pm$ 0.000 | 0.939 $\pm$ 0.000 | 0.940 $\pm$ 0.000 |
| ranolazine_mpo | 0.748 $\pm$ 0.018 | 0.735 $\pm$ 0.013 | 0.765 $\pm$ 0.029 | 0.714 $\pm$ 0.008 | 0.699 $\pm$ 0.026 |
| scaffold_hop | 0.525 $\pm$ 0.013 | 0.548 $\pm$ 0.019 | 0.521 $\pm$ 0.034 | 0.533 $\pm$ 0.012 | 0.494 $\pm$ 0.011 |
| sitagliptin_mpo | 0.194 $\pm$ 0.121 | 0.186 $\pm$ 0.055 | 0.393 $\pm$ 0.083 | 0.066 $\pm$ 0.019 | 0.363 $\pm$ 0.057 |
| thiothixene_rediscovery | 0.495 $\pm$ 0.040 | 0.559 $\pm$ 0.027 | 0.367 $\pm$ 0.027 | 0.438 $\pm$ 0.008 | 0.315 $\pm$ 0.017 |
| troglitazone_rediscovery | 0.348 $\pm$ 0.012 | 0.410 $\pm$ 0.015 | 0.320 $\pm$ 0.018 | 0.354 $\pm$ 0.016 | 0.263 $\pm$ 0.024 |
| valsartan_smarts | 0.000 $\pm$ 0.000 | 0.000 $\pm$ 0.000 | 0.000 $\pm$ 0.000 | 0.000 $\pm$ 0.000 | 0.000 $\pm$ 0.000 |
| zaleplon_mpo | 0.333 $\pm$ 0.026 | 0.221 $\pm$ 0.072 | 0.325 $\pm$ 0.027 | 0.206 $\pm$ 0.006 | 0.334 $\pm$ 0.041 |
| Sum | 13.471 | 13.156 | 13.024 | 12.223 | 12.054 |

Table 2: **Top-100 diversity, top-100 novelty, and top-100 SA score results.** The results are the average values of all 23 tasks. The best results are highlighted in bold.

| Metric | $f$-RAG (ours) | Genetic GFN | Mol GA | REINVENT |
|---|---|---|---|---|
| Average diversity $\uparrow$ | **0.532** | 0.443 | 0.491 | 0.468 |
| Average novelty $\uparrow$ | 0.800 | 0.724 | **0.845** | 0.540 |
| Average SA score $\downarrow$ | **2.026** | 3.770 | 4.605 | 3.207 |

Table 3: **Novel top 5% docking score (kcal/mol) results.** The results are the means and standard deviations of 3 independent runs. The results for RationaleRL, PS-VAE, RetMol, and GEAM are taken from Lee et al. [25]. Other baseline results except for Genetic GFN are taken from Lee et al. [24]. Lower is better, and the best results are highlighted in bold.

| Method | Target protein | | | | |
|---|---|---|---|---|---|
| | parp1 | fa7 | 5ht1b | braf | jak2 |
| JT-VAE [16] | -9.482 ± 0.132 | -7.683 ± 0.048 | -9.382 ± 0.332 | -9.079 ± 0.069 | -8.885 ± 0.026 |
| REINVENT [35] | -8.702 ± 0.523 | -7.205 ± 0.264 | -8.770 ± 0.316 | -8.392 ± 0.400 | -8.165 ± 0.277 |
| Graph GA [14] | -10.949 ± 0.532 | -7.365 ± 0.326 | -10.422 ± 0.670 | -10.789 ± 0.341 | -10.167 ± 0.576 |
| MORLD [15] | -7.532 ± 0.260 | -6.263 ± 0.165 | -7.869 ± 0.650 | -8.040 ± 0.337 | -7.816 ± 0.133 |
| HierVAE [17] | -9.487 ± 0.278 | -6.812 ± 0.274 | -8.081 ± 0.252 | -8.978 ± 0.525 | -8.285 ± 0.370 |
| GA+D [32] | -8.365 ± 0.201 | -6.539 ± 0.297 | -8.567 ± 0.177 | -9.371 ± 0.728 | -8.610 ± 0.104 |
| MARS [45] | -9.716 ± 0.082 | -7.839 ± 0.018 | -9.804 ± 0.073 | -9.569 ± 0.078 | -9.150 ± 0.114 |
| GEGL [1] | -9.329 ± 0.170 | -7.470 ± 0.013 | -9.086 ± 0.067 | -9.073 ± 0.047 | -8.601 ± 0.038 |
| RationaleRL [18] | -10.663 ± 0.086 | -8.129 ± 0.048 | -9.005 ± 0.155 | *No hit found* | -9.398 ± 0.076 |
| FREED [46] | -10.579 ± 0.104 | -8.378 ± 0.044 | -10.714 ± 0.183 | -10.561 ± 0.080 | -9.735 ± 0.022 |
| PS-VAE [20] | -9.978 ± 0.091 | -8.028 ± 0.050 | -9.887 ± 0.115 | -9.637 ± 0.049 | -9.464 ± 0.129 |
| MOOD [24] | -10.865 ± 0.113 | -8.160 ± 0.071 | -11.145 ± 0.042 | -11.063 ± 0.034 | -10.147 ± 0.060 |
| RetMol [42] | -8.590 ± 0.475 | -5.448 ± 0.688 | -6.980 ± 0.740 | -8.811 ± 0.574 | -7.133 ± 0.242 |
| GEAM [25] | -12.891 ± 0.158 | -9.890 ± 0.116 | -12.374 ± 0.036 | -12.342 ± 0.095 | -11.816 ± 0.067 |
| Genetic GFN [19] | -9.227 ± 0.644 | -7.288 ± 0.433 | -8.973 ± 0.804 | -8.719 ± 0.190 | -8.539 ± 0.592 |
| $f$-RAG (ours) | **-12.945** ± 0.053 | **-9.899** ± 0.205 | **-12.670** ± 0.144 | **-12.390** ± 0.046 | **-11.842** ± 0.316 |

synthesizability, and novelty. Following the previous works, we use docking score calculated by QuickVina 2 [2] with five protein targets, parp1, fa7, 5ht1b, braf, and jak2, to measure binding affinity. We use quantitative estimates of drug-likeness (QED) [4] and SA [8] to measure drug-likeness and synthesizability, respectively. Following the previous works, we set the target property $y$ as follows:

$$y = \widehat{\text{DS}} \times \text{QED} \times \widehat{\text{SA}} \in [0, 1], \tag{4}$$

where $\widehat{\text{DS}}$ and $\widehat{\text{SA}}$ are the normalized DS and SA according to Eq. (7) in Section D.4. We generate 3,000 molecules and evaluate them using **novel top 5% DS (kcal/mol)**, which indicates the mean DS of the top 5% unique and novel hits. Here, novel hits are defined as the molecules that satisfy all the following criteria: (the maximum similarity with the training molecules) $< 0.4$, DS $<$ (the median DS of known active molecules), QED $> 0.5$, and SA $< 5$. Further details are provided in Section D.4.

**Baselines.** **JT-VAE** [16], **HierVAE** [17], **MARS** [45], **RationaleRL** [18], **FREED** [46], **PS-VAE** [20], and **GEAM** [25] are the methods that first construct a fragment vocabulary using a molecular dataset, then learn to assemble those fragments to generate new molecules. On the other hand, **Graph GA** [14], **GEGL** [1], and **Genetic GFN** [19] are GA-based methods that utilize the information of fragments by adopting the fragment-based crossover operation. **GA+D** [32] is a discriminator-enhanced GA method that operates on the SELFIES representation. **RetMol** [42] is a retrieval-based method that uses retrieved example molecules to augment the generation of a pre-trained molecular language model. **REINVENT** [35] and **MORLD** [15] are RL models that operate on SMILES and graph molecular representations, respectively. **MOOD** [24] is a diffusion model that employs an out-of-distribution control to improve novelty.

**Results.** The results are shown in Table 3. In all five tasks, $f$-RAG outperforms all the baselines. Note that the evaluation metric, novel top 5% DS, is designed to reflect the nature of the drug discovery problem, which optimizes multiple target properties by finding a good balance between exploration and exploitation. The results demonstrate the superiority of $f$-RAG in generating drug-like, synthesizable, and novel drug candidates that have high binding affinity to the target protein with the improved exploration-exploitation trade-off. We additionally visualize the generated molecules in Figure 8.

**Additional comparison with GEAM.** To further justify the advantage of $f$-RAG over GEAM, we additionally include the results of seven multi-property optimization (MPO) tasks in the PMO benchmark (Section 4.1) in Table 7. As we can see, $f$-RAG significantly outperforms GEAM in all the tasks, validating its applicability to a wide range of drug discovery problems.

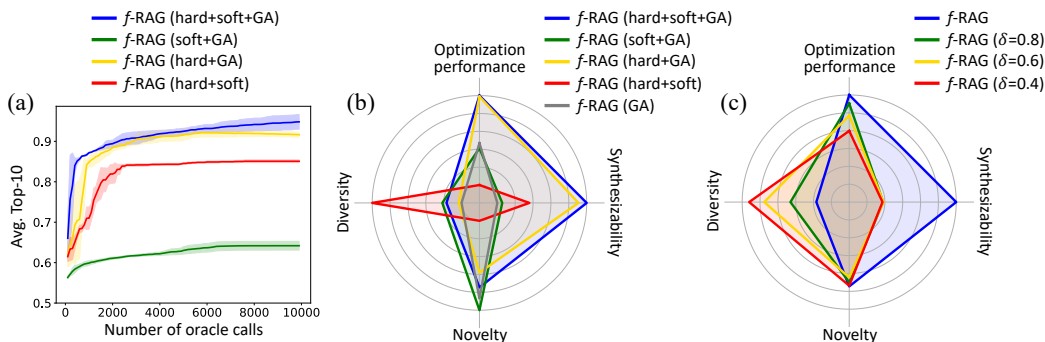

Figure 5: **(a) The optimization curves** in the deco_hop task of the PMO benchmark of the ablated $f$-RAGs. Solid lines denote the mean and shaded areas denote the standard deviation of 3 independent runs. **(b) Overall results** of the ablated $f$-RAGs. **(c) Results with different values** of $\delta$ of the similarity-based fragment filter.

## 4.3 Analysis

**Ablation study.** To verify the effect of each component of $f$-RAG, we conduct experiments with various combinations of the components, i.e., hard fragment retrieval (hard), soft fragment retrieval (soft), and genetic fragment modification (GA), in Figure 5(a) and Figure 5(b). The detailed results are shown in Table 8 and Table 9. For $f$-RAG (soft+GA) that does not utilize the hard fragment retrieval, we randomly initialized the fragment vocabularies instead of selecting the top-$N_{frag}$ fragments based on Eq. (1) and let the model use random hard fragments. Note that even though $f$-RAG (hard+soft), $f$-RAG that does not utilize the genetic fragment modification, shows very high diversity, the value is not meaningful as the generated molecules are not highly optimized, not novel, and not synthesizable. The same applies to the high novelty of $f$-RAG (soft+GA) and $f$-RAG (GA). On the contrary, the full $f$-RAG achieves the best optimization performance and synthesizability while showing reasonably high diversity and novelty. Importantly, $f$-RAG outperforms $f$-RAG (hard+GA) in all the metrics, especially for diversity and novelty, indicating the soft fragment retrieval aids the model in generating more diverse and novel drug candidates while maintaining their high target properties.

**Controlling optimization performance-diversity trade-off.** It is well-known that molecular diversity is important in drug discovery, especially when the oracle has noise. However, optimization performance against the target property and molecular diversity are conflicting factors [10, 19]. To control this trade-off between optimization performance and diversity, we additionally introduce a *similarity-based fragment filtering* strategy, which excludes similar fragments from the fragment vocabulary. Specifically, when updating the vocabulary, new fragments that have a higher Tanimoto similarity than $\delta$ to fragments in the vocabulary are filtered out. Morgan fingerprints of radius 2 and 1024 bits are used to calculate the Tanimoto similarity. The results of the similarity-based fragment filter with different values of $\delta$ are shown in Figure 5(c), verifying $f$-RAG can increase diversity at the expense of optimization performance by controlling $\delta$.

**Effect of the fragment vocabulary update.** To analyze the effect of the dynamic update of the fragment vocabulary during generation, we visualize the distribution of the docking scores of molecules generated by $f$-RAG with or with the fragment vocabulary update in Figure 6. The dynamic update allows $f$-RAG to generate more optimized molecules in terms of the target property. Notably, with the dynamic update, $f$-RAG can discover molecules that have higher binding affinity to the target protein than the top molecule in the training set, and we visualize an example of such molecules in the figure. Note that the molecule also has low similarity (0.321) with the training molecules. This result demonstrates the explorability of $f$-RAG to discover drug candidates that lie beyond the training distribution.

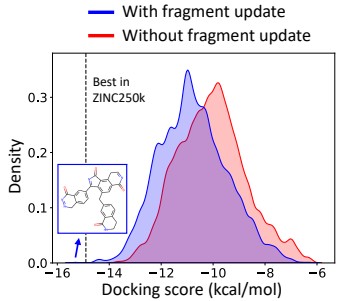

Figure 6: **DS distribution** of molecules generated by $f$-RAG with or without the fragment vocabulary update in the jak2 task.

# 5 Conclusion

We proposed $f$-RAG, a new fragment-based molecular generative framework that utilizes hard fragment retrieval, soft fragment retrieval, and genetic fragment modification. With hard fragment retrieval, $f$-RAG can explicitly exploit the information contained in existing fragments that contribute to the target properties. With soft fragment retrieval, $f$-RAG can balance between exploitation of existing chemical knowledge and exploration beyond the existing fragment vocabulary. Soft fragment retrieval with SAFE-GPT allows $f$-RAG to propose new fragments that are likely to contribute to the target chemical properties. The proposed novel fragments are then dynamically incorporated in the fragment vocabulary throughout generation. This exploration is further enhanced with genetic fragment modification, which modifies fragments with genetic operations and updates the vocabulary. Through extensive experiments, we demonstrated the efficacy of $f$-RAG to improve the exploration-exploitation trade-off. $f$-RAG outperforms previous methods, achieving state-of-the-art performance to synthesize diverse, novel, and synthesizable drug candidates with high target properties.

## Acknowledgements

We would like to thank the NVIDIA GenAIR team for their comments during the development of the framework.

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

## A    Limitations

Since our proposed $f$-RAG is built on a pre-trained backbone molecular language model, it relies heavily on the generation performance of the backbone model. However, this also means that our method delegates the difficult task of molecule generation to a large model and lets the lightweight fragment injection module take care of the relatively easy task of fragment retrieval augmentation. This strategy enables very efficient and fast training (Section D.5) and makes $f$-RAG a simple but powerful method to solve various drug discovery tasks.

## B    Broader Impacts

Through our paper, we demonstrated that $f$-RAG can achieve improved trade-offs between various considerations in drug discovery, showing its strong applicability to real-world drug discovery tasks. However, given its effectiveness, $f$-RAG has the possibility to be used maliciously to generate harmful molecules. This can be prevented by setting the target properties to comprehensively consider toxicity and other side effects, or filtering out toxic fragments from the fragment vocabulary during generation.

## C    Generation Process of $f$-RAG

---
**Algorithm 1** Generation Process of $f$-RAG

---
**Input:** Dataset $\mathcal{D}$, fragment vocabulary size $N_{\text{frag}}$, molecule population size $N_{\text{mol}}$,
        number of soft fragments $K$, number of total generations $G$,
        number of SAFE-GPT generations per cycle $G_{\text{SAFE-GPT}}$,
        number of GA generations per cycle $G_{\text{GA}}$
Set $\mathcal{V}_{\text{arm}} \leftarrow$ top-$N_{\text{frag}}$ arms obtained from $\mathcal{D}$ (Eq. (1))
Set $\mathcal{V}_{\text{linker}} \leftarrow$ top-$N_{\text{frag}}$ linkers obtained from $\mathcal{D}$ (Eq. (1))
Set $\mathcal{P} \leftarrow \emptyset$
Set $\mathcal{M} \leftarrow \emptyset$
**while** $|\mathcal{M}| < G$ **do**
  ▷ *Fragment retrieval-augmented SAFE-GPT generation*
  **for** $i = 1, 2, \ldots, N_{\text{SAFE-GPT}}$ **do**
    Randomly retrieve two hard fragments $F_{\text{hard},1}, F_{\text{hard},2}$ from $\mathcal{V}_{\text{arm}} \cup \mathcal{V}_{\text{linker}}$
    Randomly retrieve $K$ soft fragments $\{F_{\text{soft}}\}_{k=1}^{K}$ from $\mathcal{V}_{\text{arm}} \cup \mathcal{V}_{\text{linker}}$
    Using $F_{\text{hard},1}.F_{\text{hard},2}$ as input and $\{F_{\text{soft}}\}_{k=1}^{K}$ as soft fragments, run SAFE-GPT to generate a molecule $x$
    Update $\mathcal{M} \leftarrow \mathcal{M} \cup \{x\}$
    Decompose $x$ into $F_{\text{arm},1}, F_{\text{linker}}$, and $F_{\text{arm},2}$
    Update $\mathcal{V}_{\text{arm}} \leftarrow$ top-$N_{\text{frag}}$ arms from $\mathcal{V}_{\text{arm}} \cup \{F_{\text{arm},1}, F_{\text{arm},2}\}$   ⎫
    Update $\mathcal{V}_{\text{linker}} \leftarrow$ top-$N_{\text{frag}}$ linkers from $\mathcal{V}_{\text{linker}} \cup F_{\text{linker}}$   ⎬ Update
    Update $\mathcal{P} \leftarrow$ top-$N_{\text{mol}}$ from $\mathcal{P} \cup \{x\}$   ⎭
  **end for**
  ▷ *GA generation*
  **for** $i = 1, 2, \ldots, N_{\text{GA}}$ **do**
    Select parent molecules from $\mathcal{P}$
    Perform crossover and mutation to generate a molecule $x$
    Update $\mathcal{M} \leftarrow \mathcal{M} \cup \{x\}$
    Decompose $x$ into $F_{\text{arm},1}, F_{\text{linker}}$, and $F_{\text{arm},2}$
    Update $\mathcal{V}_{\text{arm}} \leftarrow$ top-$N_{\text{frag}}$ arms from $\mathcal{V}_{\text{arm}} \cup \{F_{\text{arm},1}, F_{\text{arm},2}\}$   ⎫
    Update $\mathcal{V}_{\text{linker}} \leftarrow$ top-$N_{\text{frag}}$ linkers from $\mathcal{V}_{\text{linker}} \cup F_{\text{linker}}$   ⎬ Update
    Update $\mathcal{P} \leftarrow$ top-$N_{\text{mol}}$ from $\mathcal{P} \cup \{x\}$   ⎭
  **end for**
**end while**
**Output:** Generated molecules $\mathcal{M}$

---

# D  Experimental Details

## D.1  Training of Fragment Injection Module

In this section, we describe the details for training the fragment injection module, the only part of the $f$-RAG framework that is trained. $f$-RAG has 2,362,368 trainable parameters, coming from the fragment injection module. These correspond to only 2.64% of the total parameters of 89,648,640, indicating that the fragment injection module is very lightweight compared to the backbone molecular language model.

Throughout the paper, we used the official codebase including the pre-trained SAFE-GPT[3] of Noutahi et al. [34]. Following the codebase, we used the HuggingFace Transformer library [44] (Apache-2.0 license) to train the fragment injection module. We set the number of retrieved soft fragments to $K = 10$. For the layer of the backbone language model that the fragment injection module will be inserted behind, we conducted experiments with the search space $L \in \{1, 6\}$, and found the $L = 1$-st layer works well. The results showing this comparison are provided in Table 11. The fragment injection module was trained to 8 epochs with a learning rate of $1 \times 10^{-4}$ using the AdamW optimizer [29]. We performed searches with the search spaces (epoch) $\in \{5, 8, 10\}$ and (learning rate) $\in \{1 \times 10^{-4}, 5 \times 10^{-3}\}$, respectively.

## D.2  Genetic Operations for Fragment Modification

As described in Section 3.3, we adopt the operations of Graph GA [14] to modify the generated fragments to further enhance exploration of $f$-RAG. Specifically, in the crossover operation, parents are cut at random positions at ring or non-ring positions with a probability of 50%, and random fragments from the cut are combined to generate offspring. In the mutation operation, bond insertion/deletion, atom insertion/deletion, bond order swapping, or atom changes are performed on the offspring molecule with a predefined probability.

## D.3  Experiments on PMO Benchmark

In this section, we describe the experimental details used in the experiments of Section 4.1.

**Implementation details.**  We used the official codebase[4] of Gao et al. [10] for implementing the experiments. We used ZINC250k [13] with the same train/test split used by Kusner et al. [22] to train the fragment injection module and construct the initial fragment vocabulary. We set the size of the fragment vocabulary to $N_{\text{frag}} = 50$ and the size of the molecule population to $N_{\text{mol}} = 50$. When constructing the fragment vocabulary, fragments were filtered according to the number of atoms. We searched the size range in the search space [(min. number of atoms), (max. number of atoms)] $\in \{[5, 12], [10, 16], [10, 30]\}$. As a result, we used the range $[5, 12]$ for the albuterol_similarity, isomers_c7h8n2o2m, isomers_c9h10n2o2pf2cl, median1, qed, sitagliptin_mpo, zaleplon_mpo tasks, $[10, 30]$ for the gsk3b and jnk3 tasks, and $[10, 16]$ for the rest of the tasks. We also confined the size of the generate molecules. We used the range $[10, 30]$ for the albuterol_similarity, isomers_c7h8n2o2m, isomers_c9h10n2o2pf2cl, median1, qed, sitagliptin_mpo, zaleplon_mpo tasks, $[30, 80]$ for the gsk3b and jnk3 tasks, and $[20, 50]$ for the rest of the tasks. We set the mutation rate of the GA to 0.1. We set the number of SAFE-GPT generation and number of GA generation in one cycle to $G_{\text{SAFE-GPT}} = 10$ and $G_{\text{GA}} = 10$, respectively.

**Measuring novelty and synthesizability.**  Following previous works [18, 45, 24, 25], novelty of the generated molecules $\mathcal{X}$ is measured by the fraction of molecules that have a similarity less than 0.4 compared to its nearest neighbor $x_{\text{SNN}}$ in the training set as follows:

$$\text{Novelty} = \frac{1}{N} \sum_{x \in \mathcal{X}} \mathbb{1}\{\text{sim}(x, x_{\text{SNN}}) < 0.4\}, \tag{5}$$

where $N = |\mathcal{X}|$ and $\text{sim}(x, x')$ is the pairwise Tanimoto similarity of molecules $x$ and $x'$. The similarity is calculated using Morgan fingerprints of radius 2 and 1024 bits obtained by the RDKit [23]

---

[3]https://github.com/datamol-io/safe (Apache-2.0 license)
[4]https://github.com/wenhao-gao/mol_opt (MIT license)

library. On the other hand, since synthetic accessibility (SA) [8] scores range from 1 to 10 with higher scores indicating more difficult synthesis, we measure synthesizability using the normalized SA score as follows:

$$\text{Synthesizability} = \frac{10 - \text{SA}}{9}. \tag{6}$$

**Visualizing the Radar Plots.** To draw Figure 1 and Figure 5, sum AUC top-10, average top-100 diversity, average top-100 novelty, and normalized average top-100 SA score on the PMO benchmark in Table 1 and Table 2 were used to indicate optimization performance, diversity, novelty, and synthesizability, respectively. After the SA score is normalized according to Eq. (6), all the values are min-max normalized to $[0, 1]$.

## D.4 Optimization of Docking Score under QED, SA, and Novelty Constraints

In this section, we describe the experimental details used in the experiments of Section 4.2.

**Implementation details.** We used the official codebase[5] of Lee et al. [24] for implementing the experiments. Following Lee et al. [24] and Lee et al. [25], we used ZINC250k [13] with the same train/test split used by Kusner et al. [22] to train the fragment injection module and construct the initial fragment vocabulary. As in Section D.3, when constructing the fragment vocabulary, fragments were filtered based on their number of atoms. We set the range to $[5, 12]$. To confinee the size of the generate molecules, we used the range $[20, 40]$. The mutation rate of the GA was set to $0.1$. We set the number of SAFE-GPT generation and number of GA generation in one cycle to $G_{\text{SAFE-GPT}} = 1$ and $G_{\text{GA}} = 3$, respectively.

**Measuring docking score, QED, SA, and novelty.** We strictly followed the setting used in previous works [24, 25] to measure these properties. Specifically, we used QuickVina 2 [2] to calculate docking scores (DSs). We used the RDKit [23] library to calculate QED and SA. Following previous works, we compute the normalized DS ($\widehat{\text{DS}}$) and the normalized SA ($\widehat{\text{SA}}$) as follows:

$$\widehat{\text{DS}} = -\frac{\text{clip}(\text{DS})}{20} \in [0, 1], \quad \widehat{\text{SA}} = \frac{10 - \text{SA}}{9} \in [0, 1], \tag{7}$$

where clip is the function that clips the value in the range $[-20, 0]$. As in Section D.3, novelty is determined as $\mathbb{1}\{\text{sim}(x, x_{\text{SNN}}) < 0.4\}$.

## D.5 Computing resources

We trained the fragment injection module using one GeForce RTX 3090 GPU. The training took less than 4 hours. We generated molecules using one Titan XP (12GB), GeForce RTX 2080 Ti (11GB), or GeForce RTX 3090 GPU (24GB). The experiments on each task in Section 4.1 took less than 2 hours and the experiments on each task in Section 4.2 took approximately 2 hours.

---

[5]https://github.com/SeulLee05/MOOD (MIT license)

# E  Additional Experimental Results

In this section, we provide additional experimental results.

**Diversity, novelty, and synthesizability results on PMO benchmark.** We provide the full results of Table 2 in Table 4, Table 5, and Table 6. As shown in the tables, $f$-RAG achieves the best diversity and synthesizability, while showing the second best novelty. We also provide the pairwise trade-off plots in Figure 7.

**Comparison with GEAM on PMO benchmark.** To provide additional comparisons of $f$-RAG to GEAM [25], we followed the setup of Lee et al. [25] to conduct experiments on the seven MPO tasks in the PMO benchmark. As shown in Table 7, $f$-RAG largely outperforms the baselines including GEAM in all of the tasks.

**Ablation study.** We provide the full results of Figure 5(b) in Table 8 and Table 9. As shown in the tables, Figure 5(a), and Figure 5(b), the three components of $f$-RAG, hard fragment retrieval, soft fragment retrieval, and genetic fragment modification, are essential to the superior performance of $f$-RAG and improved balance between various drug discovery considerations.

**Controlling optimization performance-diversity trade-off.** We provide the full results of Figure 5(c) in Table 10. As shown in the table and Figure 5(c), $f$-RAG can effectively control the performance-diversity trade-off with the similarity-based fragment filtering.

**Effect of location of the fragment injection module.** We provide the results of $f$-RAG with $L = 1$ and $L = 6$ in Table 11. As shown in the table, $f$-RAG with $L = 1$ works better than $f$-RAG with $L = 6$, probably because injecting information from soft fragments in an earlier layer can carry more information into the final augmented embedding. As a result of this experiment, $f$-RAG in the main experiments used $L = 1$.

**Visualization of generated molecules.** We visualize examples of the generated molecules by $f$-RAG in Figure 8. The examples are drawn randomly from the molecules used to calculate the novel top 5% DS values in Table 3.

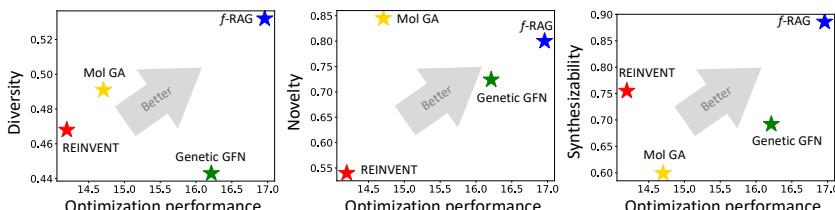

Figure 7: **Various trade-offs between the basic considerations in drug discovery.** Sum AUC top-10, average top-100 diversity, average top-100 novelty, and normalized average top-100 SA score on the PMO benchmark are used to measure optimization performance, diversity, novelty, and synthesizability, respectively.

Table 4: **Top-100 diversity results.** The results are the mean of 3 independent runs. The best results are highlighted in bold.

| Oracle | $f$-RAG (ours) | Genetic GFN | Mol GA | REINVENT |
|---|---|---|---|---|
| albuterol_similarity | 0.505 | 0.375 | **0.596** | 0.418 |
| amlodipine_mpo | **0.394** | 0.280 | 0.371 | 0.334 |
| celecoxib_rediscovery | **0.360** | 0.283 | 0.318 | 0.306 |
| deco_hop | 0.482 | **0.536** | 0.422 | 0.445 |
| drd2 | 0.491 | 0.521 | **0.548** | 0.496 |
| fexofenadine_mpo | **0.526** | 0.424 | 0.508 | 0.403 |
| gsk3b | 0.329 | 0.159 | 0.085 | **0.402** |
| isomers_c7h8n2o2 | **0.824** | 0.786 | 0.820 | 0.765 |
| isomers_c9h10n2o2pf2cl | **0.797** | 0.621 | 0.789 | 0.699 |
| jnk3 | 0.318 | **0.354** | 0.045 | 0.205 |
| median1 | **0.614** | 0.430 | 0.543 | 0.436 |
| median2 | 0.424 | **0.470** | 0.424 | 0.382 |
| mestranol_similarity | **0.572** | 0.214 | 0.443 | 0.322 |
| osimertinib_mpo | 0.480 | 0.450 | **0.505** | 0.444 |
| perindopril_mpo | 0.375 | 0.394 | **0.454** | 0.346 |
| qed | **0.816** | 0.642 | 0.601 | 0.699 |
| ranolazine_mpo | **0.570** | 0.383 | 0.420 | 0.318 |
| scaffold_hop | **0.479** | 0.465 | 0.451 | 0.456 |
| sitagliptin_mpo | 0.647 | 0.165 | 0.632 | **0.673** |
| thiothixene_rediscovery | 0.428 | 0.388 | **0.438** | 0.387 |
| troglitazone_rediscovery | **0.445** | 0.391 | 0.407 | 0.410 |
| valsartan_smarts | 0.600 | 0.881 | **0.902** | 0.882 |
| zaleplon_mpo | **0.751** | 0.581 | 0.563 | 0.530 |
| Average | **0.532** | 0.443 | 0.491 | 0.468 |

Table 5: **Top-100 novelty results.** The results are the mean of 3 independent runs. The best results are highlighted in bold.

| Oracle | $f$-RAG (ours) | Genetic GFN | Mol GA | REINVENT |
|---|---|---|---|---|
| albuterol_similarity | 0.543 | 0.667 | **0.957** | 0.553 |
| amlodipine_mpo | 0.000 | 0.037 | **0.113** | 0.000 |
| celecoxib_rediscovery | 0.930 | **0.980** | 0.885 | 0.943 |
| deco_hop | 0.650 | 0.923 | **1.000** | 0.893 |
| drd2 | 0.570 | 0.780 | **0.987** | 0.357 |
| fexofenadine_mpo | **1.000** | 0.940 | **1.000** | 0.547 |
| gsk3b | **1.000** | **1.000** | **1.000** | 0.757 |
| isomers_c7h8n2o2 | **0.953** | 0.647 | 0.947 | 0.373 |
| isomers_c9h10n2o2pf2cl | 0.993 | 0.747 | **1.000** | 0.273 |
| jnk3 | **1.000** | 0.933 | **1.000** | **1.000** |
| median1 | 0.437 | 0.810 | **0.980** | 0.433 |
| median2 | 0.880 | 0.127 | **0.943** | 0.010 |
| mestranol_similarity | 0.850 | 0.893 | **0.970** | 0.923 |
| osimertinib_mpo | 0.917 | 0.737 | **1.000** | 0.617 |
| perindopril_mpo | 0.973 | 0.850 | **1.000** | 0.930 |
| qed | **0.757** | 0.423 | 0.053 | 0.063 |
| ranolazine_mpo | **1.000** | 0.453 | **1.000** | 0.290 |
| scaffold_hop | 0.810 | 0.993 | **1.000** | 0.930 |
| sitagliptin_mpo | 0.920 | **1.000** | **1.000** | 0.303 |
| thiothixene_rediscovery | 0.927 | **0.980** | 0.960 | 0.920 |
| troglitazone_rediscovery | 0.683 | **0.917** | 0.683 | 0.833 |
| valsartan_smarts | **0.820** | 0.200 | 0.323 | 0.123 |
| zaleplon_mpo | **0.793** | 0.610 | 0.750 | 0.343 |
| Average | 0.800 | 0.724 | **0.845** | 0.540 |

Table 6: **Top-100 SA score results.** The results are the mean of 3 independent runs. Lower is better and the best results are highlighted in bold.

| Oracle | $f$-RAG (ours) | Genetic GFN | Mol GA | REINVENT |
|---|---|---|---|---|
| albuterol_similarity | **1.451** | 3.241 | 3.822 | 3.311 |
| amlodipine_mpo | **2.218** | 3.675 | 3.806 | 3.645 |
| celecoxib_rediscovery | **1.584** | 2.735 | 2.388 | 2.667 |
| deco_hop | **2.601** | 4.333 | 5.066 | 3.262 |
| drd2 | **1.079** | 2.605 | 3.051 | 2.514 |
| fexofenadine_mpo | **1.837** | 4.437 | 4.721 | 4.193 |
| gsk3b | **1.253** | 4.785 | 10.000 | 2.964 |
| isomers_c7h8n2o2 | **1.997** | 3.354 | 4.721 | 3.323 |
| isomers_c9h10n2o2pf2cl | **1.125** | 4.108 | 5.314 | 2.828 |
| jnk3 | **1.746** | 4.679 | 10.000 | 4.202 |
| median1 | **2.036** | 4.098 | 4.516 | 4.058 |
| median2 | 3.408 | 3.022 | 3.861 | **2.957** |
| mestranol_similarity | **1.987** | 4.241 | 4.780 | 4.131 |
| osimertinib_mpo | **3.053** | 3.734 | 4.293 | 3.257 |
| perindopril_mpo | **2.153** | 4.674 | 4.648 | 4.146 |
| qed | 2.968 | 3.340 | 2.501 | **2.232** |
| ranolazine_mpo | **2.055** | 3.489 | 4.422 | 3.096 |
| scaffold_hop | 3.691 | 3.513 | 5.146 | **3.030** |
| sitagliptin_mpo | **1.627** | 5.652 | 5.479 | 2.912 |
| thiothixene_rediscovery | 2.866 | 2.621 | 2.872 | **2.444** |
| troglitazone_rediscovery | **1.631** | 4.541 | 3.672 | 3.255 |
| valsartan_smarts | **2.867** | 3.127 | 3.530 | 3.100 |
| zaleplon_mpo | **1.574** | 2.717 | 3.296 | 2.236 |
| Average | **2.026** | 3.770 | 4.605 | 3.207 |

Table 7: **PMO MPO AUC top-100 results.** The results are the means of 3 runs. The results for REINVENT, Graph GA, and SELFIES-REINVENT are taken from Gao et al. [10], and the results for GEAM are taken from Lee et al. [25]. The best results are highlighted in bold.

| Method | MPO Benchmark | | | | | | |
|---|---|---|---|---|---|---|---|
| | Amlodipine | Fexofenadine | Osimertinib | Perindopril | Ranolazine | Sitagliptin | Zaleplon |
| REINVENT [35] | 0.608 | 0.752 | 0.806 | 0.511 | 0.719 | 0.006 | 0.325 |
| Graph GA [14] | 0.622 | 0.731 | 0.799 | 0.503 | 0.670 | 0.330 | 0.305 |
| SELFIES-REINVENT | 0.574 | 0.705 | 0.791 | 0.487 | 0.695 | 0.118 | 0.257 |
| GEAM [25] | 0.626 | 0.799 | 0.831 | 0.514 | 0.714 | 0.417 | 0.402 |
| $f$-RAG (ours) | **0.704** $\pm$ 0.018 | **0.827** $\pm$ 0.010 | **0.850** $\pm$ 0.007 | **0.640** $\pm$ 0.017 | **0.779** $\pm$ 0.016 | **0.458** $\pm$ 0.024 | **0.423** $\pm$ 0.001 |

Table 8: **PMO AUC top-10 results of the ablated versions of $f$-RAG.** The results are the mean of 3 independent runs. The best results are highlighted in bold.

| | $f$-RAG | | | | |
|---|---|---|---|---|---|
| Hard retrieval | ✓ | x | ✓ | ✓ | x |
| Soft retrieval | ✓ | ✓ | x | ✓ | x |
| GA | ✓ | ✓ | ✓ | x | ✓ |
| albuterol_similarity | **0.977** | 0.907 | 0.974 | 0.875 | 0.929 |
| amlodipine_mpo | **0.749** | 0.620 | 0.747 | 0.629 | 0.666 |
| celecoxib_rediscovery | **0.778** | 0.524 | 0.748 | 0.505 | 0.695 |
| deco_hop | **0.936** | 0.621 | 0.908 | 0.811 | 0.645 |
| drd2 | 0.992 | 0.987 | **0.993** | **0.993** | 0.976 |
| fexofenadine_mpo | **0.856** | 0.823 | 0.838 | 0.756 | 0.830 |
| gsk3b | **0.969** | 0.825 | 0.967 | 0.876 | 0.822 |
| isomers_c7h8n2o2 | **0.955** | 0.940 | 0.931 | 0.891 | 0.968 |
| isomers_c9h10n2o2pf2cl | 0.850 | 0.795 | **0.880** | 0.727 | 0.896 |
| jnk3 | **0.904** | 0.834 | 0.903 | 0.800 | 0.612 |
| median1 | **0.340** | 0.293 | 0.339 | 0.290 | 0.285 |
| median2 | **0.323** | 0.250 | 0.315 | 0.255 | 0.284 |
| mestranol_similarity | 0.671 | 0.650 | **0.683** | 0.566 | 0.550 |
| osimertinib_mpo | 0.866 | 0.835 | **0.867** | 0.821 | 0.838 |
| perindopril_mpo | **0.681** | 0.530 | 0.667 | 0.585 | 0.564 |
| qed | 0.939 | 0.937 | 0.941 | 0.937 | **0.942** |
| ranolazine_mpo | **0.820** | 0.800 | 0.809 | 0.726 | 0.808 |
| scaffold_hop | 0.576 | 0.556 | **0.589** | 0.509 | 0.522 |
| sitagliptin_mpo | 0.601 | 0.502 | **0.615** | 0.304 | 0.576 |
| thiothixene_rediscovery | **0.584** | 0.458 | 0.579 | 0.432 | 0.545 |
| troglitazone_rediscovery | 0.448 | 0.397 | **0.459** | 0.313 | 0.396 |
| valsartan_smarts | **0.627** | 0.685 | 0.624 | 0.000 | 0.540 |
| zaleplon_mpo | 0.486 | 0.462 | **0.510** | 0.447 | 0.506 |
| Sum | **16.928** | 15.231 | 16.890 | 14.048 | 15.395 |

Table 9: **Top-100 diversity, top-100 novelty, and top-100 SA score results of the ablated versions of $f$-RAG.** The results are the average values of all 23 tasks. The best results are highlighted in bold.

| | $f$-RAG | | | | |
|---|---|---|---|---|---|
| Hard retrieval | ✓ | x | ✓ | ✓ | x |
| Soft retrieval | ✓ | ✓ | x | ✓ | x |
| GA | ✓ | ✓ | ✓ | x | ✓ |
| Average diversity ↑ | 0.532 | 0.544 | 0.497 | **0.744** | 0.489 |
| Average novelty ↑ | 0.800 | **0.924** | 0.720 | 0.438 | 0.860 |
| Average SA score ↓ | **2.026** | 4.113 | 2.231 | 3.443 | 4.235 |

Table 10: **Top-100 diversity, top-100 novelty, and top-100 SA score results with different values of $\delta$ of the similarity-based fragment filter.** The results are the average values of all 23 tasks.

| Metric | $f$-RAG | $f$-RAG ($\delta = 0.8$) | $f$-RAG ($\delta = 0.6$) | $f$-RAG ($\delta = 0.4$) |
|---|---|---|---|---|
| Sum AUC ↑ | 16.928 | 16.648 | 16.262 | 15.765 |
| Average diversity ↑ | 0.532 | 0.606 | 0.681 | 0.724 |
| Average novelty ↑ | 0.800 | 0.778 | 0.751 | 0.796 |
| Average SA score ↓ | 2.026 | 3.836 | 3.825 | 3.852 |

Table 11: **Effect of location of the fragment injection module** $L$. The results are the mean of 3 independent runs. $f$-RAG in the main experiments used $L = 1$. The best results are highlighted in bold.

| Oracle | $f$-RAG ($L = 1$) | $f$-RAG ($L = 6$) |
|---|---|---|
| albuterol_similarity | **0.977** | 0.971 |
| amlodipine_mpo | **0.749** | 0.721 |
| celecoxib_rediscovery | **0.778** | 0.764 |
| deco_hop | **0.936** | 0.935 |
| drd2 | **0.992** | 0.991 |
| fexofenadine_mpo | **0.856** | 0.847 |
| gsk3b | 0.969 | **0.974** |
| isomers_c7h8n2o2 | **0.955** | 0.950 |
| isomers_c9h10n2o2pf2cl | **0.850** | 0.848 |
| jnk3 | **0.904** | 0.901 |
| median1 | 0.340 | 0.351 |
| median2 | **0.323** | 0.322 |
| mestranol_similarity | 0.671 | **0.685** |
| osimertinib_mpo | **0.866** | 0.865 |
| perindopril_mpo | 0.681 | **0.698** |
| qed | **0.939** | **0.939** |
| ranolazine_mpo | **0.820** | 0.798 |
| scaffold_hop | 0.576 | **0.586** |
| sitagliptin_mpo | **0.601** | 0.547 |
| thiothixene_rediscovery | **0.584** | 0.583 |
| troglitazone_rediscovery | **0.448** | 0.447 |
| valsartan_smarts | 0.627 | **0.685** |
| zaleplon_mpo | **0.486** | **0.486** |
| Sum | **16.928** | 16.894 |

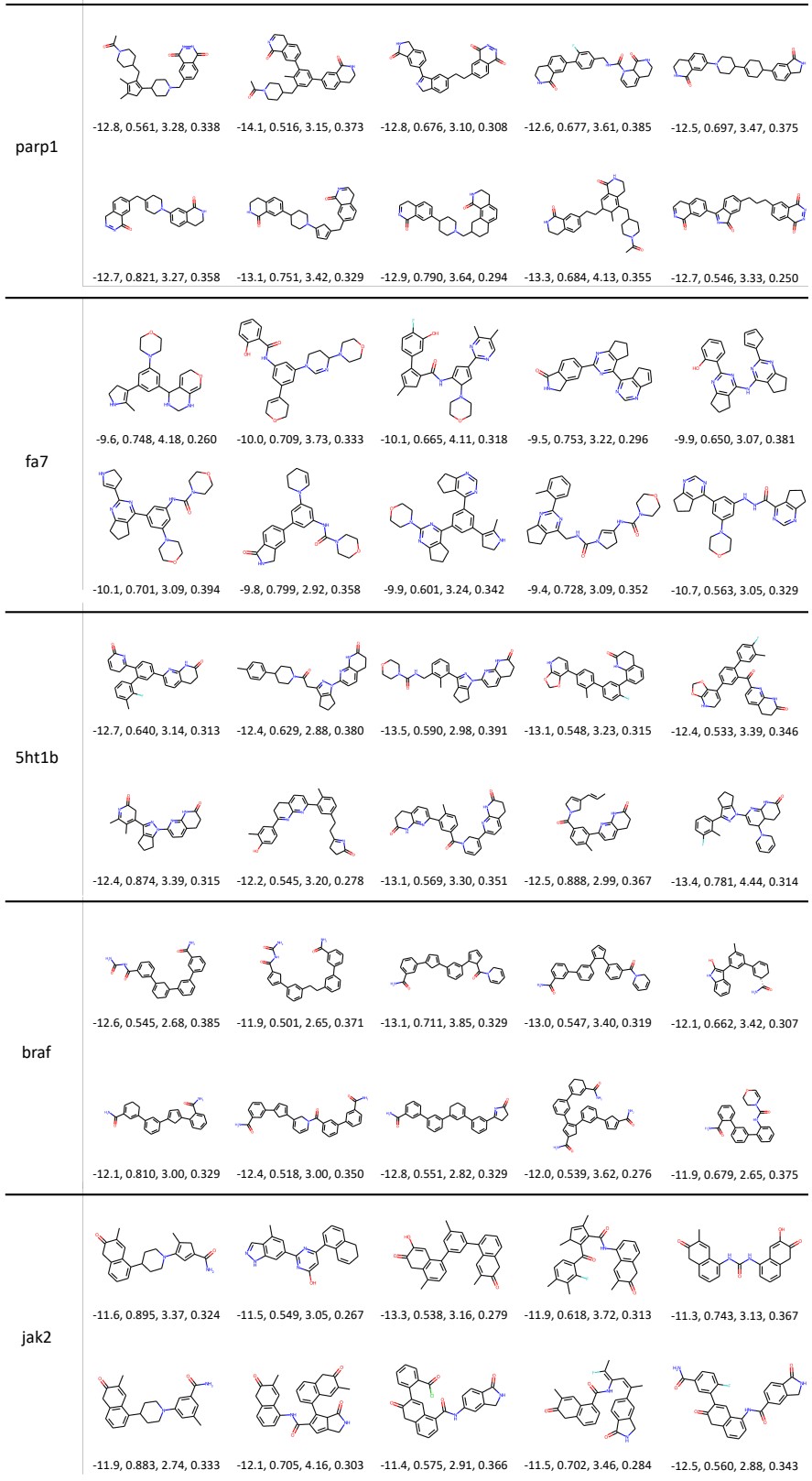

Figure 8: **The generated molecules by $f$-RAG.** Random top 5% novel hits generated in the experiments of Section 4.2 are visualized. The docking score (kcal/mol), QED, SA, and the maximum similarity with the molecules in the training set are at the bottom of each molecule.

