# OpenReview forum: "Molecule Generation with Fragment Retrieval Augmentation"
_NeurIPS.cc/2024/Conference — NeurIPS 2024 poster_

### Official Review · Reviewer_U6dy · 2024-07-02

**Soundness:** 3
**Presentation:** 3
**Contribution:** 3
**Rating:** 7
**Confidence:** 4

**Summary:**

The paper introduces a novel fragment-based molecule generation framework called Fragment Retrieval-Augmented Generation (f-RAG). This framework aims to address the limitations of existing fragment-based molecule generation methods, which often struggle to explore beyond the existing fragments in their databases.
f-RAG utilizes a pre-trained molecular generative model to propose additional fragments from input fragments, completing and generating a new molecule. It retrieves two types of fragments: hard fragments, which are explicitly included in the newly generated molecule, and soft fragments, which serve as references to guide the generation of new fragments through a trainable fragment injection module.
To further extrapolate beyond the existing fragments, f-RAG updates the fragment vocabulary with generated fragments through an iterative refinement process. This process is enhanced with post-hoc genetic fragment modification, allowing f-RAG to maintain a pool of fragments and expand it with novel and high-quality fragments through a strong generative prior. This approach enables f-RAG to achieve an improved exploration-exploitation trade-off in fragment-based molecule generation.

**Strengths:**

1. f-RAG uses a pre-trained molecular generative model to propose new fragments, allowing new fragments to be generated
2. The approach retrieves both hard fragments that are directly incorporated into the new molecule and soft fragments that guide the generation process, enhancing the diversity and effectiveness of the generated molecules.
3. f-RAG updates the fragment vocabulary with generated fragments through an iterative process, continuously improving the quality and novelty of the fragments used for molecule generation.

**Weaknesses:**

1. The improvement of f-RAG compared to Genetic GFN is not considered large enough. In many tasks, Genetic GFN actually performs much better than f-RAG, which raises the concern of generalization of the proposed method.
2. Some notations are not clear enough. For example, in Table 2, arrows could be used to illustrate lower-better or higher-better.

**Questions:**

1. Could the authors explain the potential reason that f-RAG falls behind genetic GFN?

**Limitations:**

Yes, the limitation is well-discussed.

---

> ### Author Rebuttal · Authors · 2024-08-06
>
> We sincerely thank you for your comments. We appreciate your positive comments that our paper proposes an effective strategy to enhance the quality, diversity and novelty of molecules. We address your concerns and questions below.
>
> ---
>
> **Comment 1**
>
> The improvement of f-RAG compared to Genetic GFN is not large enough. Could the authors explain the potential reason that f-RAG falls behind genetic GFN?
>
> **Response 1**
>
> Drug discovery is a comprehensive problem and **optimization performance alone is not meaningful without consideration of other factors such as diversity, novelty, and synthesizability**. Genetic GFN [1] showed competitive results with $f$-RAG in terms of optimization performance (Table 1), but significantly worse results compared to $f$-RAG in terms of diversity, novelty, and synthesizability (Figure 1, Table 2 and Figure 7). As described in Lines 232-235, the essential considerations in drug discovery often conflict with each other, and the results of Genetic GFN illustrate these trade-offs. On the contrary, our proposed **$f$-RAG effectively improves these trade-offs** by utilizing existing fragments while dynamically updating the fragment vocabulary.
>
> To further compare $f$-RAG and Genetic GFN, we evaluated the performance of Genetic GFN in the experiments of Section 4.2 in the below table. $f$-RAG outperforms Genetic GFN by a very large margin, again demonstrating its superiority as a universal method applicable to various drug discovery tasks. We will include this result in the revised paper.
>
> **Table: Novel top 5% docking score (kcal/mol) results.** The results are the means and standard deviations of 3 independent runs. Lower is better.
> | Method | parp1 | fa7 | 5ht1b | braf | jak2 |
> | --- | --- | --- | --- | --- | --- |
> | Genetic GFN | -9.227 ± 0.644 | -7.288 ± 0.433 | -8.973 ± 0.804 | -8.719 ± 0.190 | -8.539±0.592 |
> | $f$-RAG (ours) | **-12.945** ± 0.053 | **-9.899** ± 0.205 | **-12.670** ± 0.144 | **-12.390** ± 0.046 | **-11.842** ± 0.316 |
>
> ---
>
> **Comment 2**
>
> In Table 2, arrows could be used to illustrate lower-better or higher-better.
>
> **Response 2**
>
> We appreciate the suggestion to improve the readability of our paper. We will include the arrows in the revised paper.
>
> ---
>
> **References**
>
> [1] Kim et al., Genetic-guided gflownets: Advancing in practical molecular optimization benchmark, arXiv, 2024.

---

> > ### Comment · Reviewer_U6dy · 2024-08-13
> > **Reviewer Response**
> >
> > Thanks for clarifying my concerns. Considering that my score is already supportive enough, I decide to maintain my scores. Have a good luck~

---

### Official Review · Reviewer_7DmF · 2024-07-03

**Soundness:** 2
**Presentation:** 2
**Contribution:** 1
**Rating:** 3
**Confidence:** 5

**Summary:**

The paper proposed a fragment retrieval-augmented generation for molecule discovery, namely f-RAG. f-RAG retrieves two types of fragments, i.e., hard fragments and soft fragments. Hard fragments serve as build blocks that are explicitly included in the newly generated molecules, while soft fragments guide the generation of new fragments through a trainable module.

**Strengths:**

The work is well written.

The experiments show the effectiveness of the proposed method.

**Weaknesses:**

1. The novelty of the work is limited. Very similar works have been published, such as [1], [2], and [3]. They basically follow the same pipeline, which generates new molecules through LLMs, GAs, or both. Compared to these works, this work does not seem to bring new insights.

2. Moreover, the so-called soft and hard fragments are not new either. In reference [1], retrieved exemplar molecules are used as inputs to guide the generation of new molecules through trainable networks. Therefore, the concept of soft fragments is not novel.

3. In addition, reference [2] also used GAs and LLMs to generate molecules. The author needs to further clarify the differences between the two works.

4. The baselines used by the authors are not the latest. The authors should consider incorporating LLMs related molecular generation methods into the comparison.

5. Why did the author use SAFE-GPT instead of other chemical language models? Can this method be extended to other chemical large language models?

6. Some key details in genetic algorithms are missing, such as how to use mutation and crossover to generate new molecules.

[1] Wang, Zichao, et al. "Retrieval-based controllable molecule generation.", ICLR, 2023.
[2] Lee, Seul, et al. "Drug Discovery with Dynamic Goal-aware Fragments." Forty-first International Conference on Machine Learning, 2023
[3] Wang, Haorui, et al. "Efficient Evolutionary Search Over Chemical Space with Large Language Models." arXiv preprint arXiv:2406.16976 (2024).

**Questions:**

See weakness.

**Limitations:**

No potential negative societal impact

---

> ### Author Rebuttal · Authors · 2024-08-06
>
> We sincerely thank you for your comments. We appreciate your positive comments that our paper is well-written and the experiments show the effectiveness of our method. We address your concerns below.
>
> ---
>
> **Comment 1**
>
> The novelty of the work is limited compared to [1,2,3]. In [1], retrieved exemplar molecules are used as inputs to guide the generation of new molecules through trainable networks. [2] also used GAs and LLMs.
>
> **Response 1**
>
> Compared to RetMol [1], $f$-RAG (1) retrieves fragments instead of molecules, (2) utilizes two types of retrieval--hard and soft, and (3) generates molecules in a one-shot manner instead of multiple iterations. For a more detailed explanation of the differences, please see *Global Rebuttal*.
>
> Compared to GEAM [2], **$f$-RAG is better at exploring chemical space**. GEAM does not use an LLM. GEAM uses an RL model that only reassembles the given fragments, therefore unable to generate novel molecular structure and solely relies on the modification of the GA to generate new fragments. On the contrary, $f$-RAG applies fragment-level retrieval augmentation to an LLM to propose novel, high-quality fragments rather than simply reassembling existing fragments, greatly enhancing exploration beyond known fragments. Due to these differences, $f$-RAG outperforms GEAM by a large margin (Table 3 and Table 7).
>
> Compared to MolLEO [3], **$f$-RAG applies RAG to an LLM, therefore is better at generating high-quality drug candidates**. MolLEO does not use RAG and is therefore less effective at utilizing chemical knowledge. In addition, MolLEO focuses on replacing one of the crossover or mutation of Graph GA [4] by LLM-guided operations, so it is basically a GA and suffers from the common limitation of GAs--low diversity [5]. On the contrary, $f$-RAG shows an improved balance between optimization performance, diversity, novelty, and synthesizability through the proposed RAG with the fragment injection module (Figure 1). Also, we would like to kindly inform you that **this work was released in arXiv on June 23, 2024, after the NeurIPS submission deadline, and thus should be considered as a concurrent work**.
>
> ---
>
> **Comment 2**
>
> The baselines used by the authors are not the latest. The authors should consider incorporating LLM-related molecular generation methods into the comparison.
>
> **Response 2**
>
> We used extensive drug discovery tasks--23 PMO benchmark [5] tasks and 5 docking score tasks--that simulate various real-world drug discovery scenarios for a solid evaluation. In these tasks, **we compared $f$-RAG against a large number of baselines**. In Section 4.1, we employed the top-7 methods reported by the PMO benchmark and two latest SOTA methods, Genetic GFN and Mol GA, as our baselines (Line 213). Since there are 25 baselines reported in the PMO paper, this is equivalent to showing superiority of $f$-RAG over 27 baselines. In Section 4.2, we compared $f$-RAG with 14 baselines.
>
> For LLM-related methods, Wang et al. [3] was released in arXiv **after** the NeurIPS submission deadline and there is no publicly released codebase. RetMol [1] uses an LLM (a BART model) and is included as our baseline. We have made every effort to faithfully demonstrate the superiority of $f$-RAG through extensive experiments, and if you kindly suggest any strong baselines, we would be happy to compare $f$-RAG to them to make our experiments more robust.
>
> ---
>
> **Comment 3**
>
> Why did the author use SAFE-GPT instead of other chemical language models?
>
> **Response 3**
>
> We adopted Sequential Attachment-based Fragment Embedding (SAFE), non-canonical SMILES that represents molecules as a sequence of fragments, as the molecular representation. We chose SAFE because it is **well-suited for fragment-based molecule generation**, and it enables $f$-RAG to easily include the hard fragments in a generated molecule by simply providing them as an input sequence to SAFE-GPT to complete the rest of the sequence. Other language models besides GPT are also compatible with the $f$-RAG framework as long as they are trained on a SAFE dataset, and SAFE-GPT is chosen as an example since the pre-training of chemical language models is not the main interest of our paper. We emphasize that our proposed strategy of combining fragment retrieval from a goal-aware vocabulary with the SAFE representation to build a molecular optimization framework is simple but powerful, demonstrating its effectiveness in a wide range of drug discovery tasks.
>
> ---
>
> **Comment 4**
>
> Some key details in genetic algorithms are missing.
>
> **Response 4**
>
> As described in Line 181, we adopted the mutation/crossover of Graph GA [4] to further improve exploration in the chemical space. We appreciate your comment and will include the details about the genetic operations in the revised paper for the completeness of the paper. Specifically, in the crossover operation, parents are cut at random positions at ring or non-ring positions with a probability of 50%, and random fragments from the cut are combined to generate offspring. In the mutation operation, bond insertion/deletion, atom insertion/deletion, bond order swapping, or atom changes are performed on the offspring molecule with a predefined probability.
>
> **We hope our response addresses your concerns and that you consider upgrading your rating. We are happy to elaborate further if there are any remaining concerns.**
>
> ---
> **References**
>
> [1] Wang et al., Retrieval-based controllable molecule generation, ICLR, 2023.
>
> [2] Lee et al., Drug discovery with dynamic goal-aware fragments, ICML, 2024.
>
> [3] Wang et al., Efficient evolutionary search over chemical space with large language models, arXiv, 2024.
>
> [4] Jensen, A graph-based genetic algorithm and generative model/monte carlo tree search for the exploration of chemical space. Chemical science, 10(12):3567-3572, 2019.
>
> [5] Gao et al., Sample efficiency matters: a benchmark for practical molecular optimization, NeurIPS Datasets and Benchmarks, 2022.

---

### Official Review · Reviewer_SbUr · 2024-07-07

**Soundness:** 3
**Presentation:** 3
**Contribution:** 3
**Rating:** 6
**Confidence:** 3

**Summary:**

The paper introduces f-RAG, a novel framework for fragment-based molecular generation that integrates hard and soft fragment retrieval and genetic fragment modification. It aims to improve the exploration-exploitation trade-off in drug discovery by leveraging existing molecular fragments and exploring beyond the existing chemical space. On each generation, two hard fragments are sampled which will be ensured to appear in the generated molecule. Then several soft fragments are sampled to derive an embedding via a pretrained chemical language model as the guidance for generation. Another round of genetic algorithm is also implemented to further explore the neighborhood of the generated high-scoring molecules. The authors have conducted extensive experiments on various molecular optimization tasks, demonstrating f-RAG's effectiveness in generating molecules with improved optimization performance, diversity, novelty, and synthesizability.

**Strengths:**

1. The paper tries to tackle an important problem of fragment-based molecular generation, namely exploring further chemical spaces beyond known fragments. During generations, novel and high-scoring fragments will be dynamically updated into the vocabulary so that this kind of "novelty" will be passed on to further generations, enlarging the explorable chemical space.

2. The retrieval-augmented generation provide explainability to some extent, therefore has some ensurance on the quality of the generated molecules.

**Weaknesses:**

1. Integrating both language models and genetic algorithms into generation might result in huge computational cost, which will affect the practical applications of generative models. The authors should discuss and analyze the computational efficiency as well as its trade-off with the performance.

2. Commonly the RAG process will retrieve related information from large-scale database (vocabulary) through some techniques (e.g. Vectorize the data and implement query-key matching). However, in the proposed method, both the hard and soft fragments are randomly sampled from the vocabulary, which leaves great burdens on the construction of high-quality and relevant databases for each task or property.

3. The implementation details are missing, especially for the baselines which incorporate genetic algorithms, whose performance is closely related to the values of parameters like population size and number of cycles.

**Questions:**

1. Why can all the molecules be decomposed into an arm-linker-arm form? Does this formalization impose additional biases on the chemical space which can be explored? For example, what will happen if I want to generate benzene with small substituents (e.g. toluene).

**Limitations:**

The property-related fragment vocabulary can only be built if the property score can be decomposed into fragment-level sums, which limits the applications of the proposed method.

---

> ### Author Rebuttal · Authors · 2024-08-06
>
> We sincerely thank you for your comments. We appreciate your positive comments that our dynamic vocabulary update strategy expands the explorable chemical space and that our retrieval-augmented generation strategy provides chemical explainability. We address your concerns and questions below.
>
> ---
>
> **Comment 1**
>
> Integrating both language models and genetic algorithms into generation might result in huge computational costs. The authors should analyze the computational efficiency.
>
> **Response 1**
>
> First of all, we note that **GAs generally do not require GPUs and are very fast to execute**. For example, the GA adopted in our paper, Graph GA, takes only 3 minutes for a single run in the PMO benchmark (Table 7 in the benchmark paper [1]), while $f$-RAG takes 1~2 hours as described in Section D.4. During generation with $f$-RAG, the fragment retrieval augmentation and GA parts are very fast to run, and the forward pass of the backbone language model takes up most of the runtime. The slow runtime is a common limitation of autoregressive models, including the backbone model used in our paper, SAFE-GPT. Here, SAFE-GPT was chosen as an example because the pre-training of the backbone model is not the main interest of our paper, and the use of a non-autoregressive chemical language model can improve the runtime of the $f$-RAG framework.
>
> Furthermore, we also emphasize that the runtime of $f$-RAG of 1~2 hours to generate 10,000 drug candidates is **sufficiently fast to be practical for real-world drug discovery problems, especially given its effectiveness in a wide range of drug discovery tasks**.
>
> ---
>
> **Comment 2**
>
> Both the hard and soft fragments are randomly sampled from the vocabulary, which leaves great burdens on the construction of high-quality and relevant databases for each task or property.
>
> **Response 2**
>
> Instead of querying for relevant fragments every time a new molecule is generated, we proposed to construct an initial high-quality fragment vocabulary only once before generation. This strategy of confining the pool (or vocabulary) itself and randomly retrieving information is also commonly used in GAs. Through extensive experiments, we have demonstrated **the initial vocabulary construction procedure based on the target property (Eq. (1)) is very simple yet effective, and universally applicable to any task or target property**. Moreover, we proposed to dynamically refine the initial vocabulary with newly proposed fragments during generation. This gives $f$-RAG the ability to **explore beyond the best of the database**, as shown in Figure 6.
>
> ---
>
> **Comment 3**
>
> The implementation details are missing, especially for the baselines which incorporate genetic algorithms.
>
> **Response 3**
>
> For the experiments in Section 4.1 and Section 4.2, we did not reimplement any of the baselines. For the baselines in Section 4.1, the results of Genetic GFN [1] and Mol GA [2] are taken from the respective original papers and the results of other baselines are taken from the PMO benchmark paper [3]. For the baselines in Section 4.2, the results of RationaleRL, PS-VAE, RetMol, and GEAM are taken from Lee et al. [4] and the results of other baselines are taken from Lee et al. [5]. Implementation details for our $f$-RAG are included in Section D.2 and Section D.3.
>
> ---
>
> **Comment 4**
>
> Why can all the molecules be decomposed into an arm-linker-arm form?
>
> **Response 4**
>
> As described in Lines 117-118, arms are defined as fragments that have one attachment point and linkers are defined as fragments that have two attachment points in our paper. Therefore, all molecules with two or more breakable bonds (i.e., non-ring bonds in the arm-linker-arm slicing algorithm of Noutahi et al [6] we adopted) can be decomposed into arm-linker-arm forms, and we ignored the small number of molecules in the training set that cannot be decomposed. We will include this detail in the revised paper. However, we would like to mention that any other molecular decomposition algorithm is equally compatible with our $f$-RAG framework.
>
> ---
>
> **Comment 5**
>
> The property-related fragment vocabulary can only be built if the property score can be decomposed into fragment-level sums.
>
> **Response 5**
>
> **The initial vocabulary construction procedure is based on the target property at the molecular level**, and does not require the property score to be decomposed to the fragment level. $y$ in Eq. (1) is the target property value of the whole molecule, not a fragment. As explained in Lines 124-126, this scoring function evaluates the contribution of a given fragment to the target property of the molecule of which it is a part. We emphasize that the proposed fragment vocabulary construction scheme is very simple and universally applicable to any target property.
>
> ---
>
> **References**
>
> [1] Kim et al., Genetic-guided gflownets: Advancing in practical molecular optimization benchmark, arXiv, 2024.
>
> [2] Tripp et al., Genetic algorithms are strong baselines for molecule generation, arXiv, 2023.
>
> [3] Gao et al., Sample efficiency matters: a benchmark for practical molecular optimization, NeurIPS Datasets and Benchmarks, 2022.
>
> [4] Lee et al., Drug discovery with dynamic goal-aware fragments, ICML, 2024.
>
> [5] Lee et al., Exploring chemical space with score-based out-of- distribution generation, ICML, 2023.
>
> [6] Noutahi et al., Gotta be safe: a new framework for molecular design, Digital Discovery, 3(4):796–804, 2024.

---

> > ### Comment · Reviewer_SbUr · 2024-08-12
> > **Thanks for the response**
> >
> > Thank the authors for the detailed response, which has alleviated some of my concerns. I would like to maintain my recommendation of weak acceptance, and hope the authors good luck in addressing the concerns of the other reviewers.

---

### Official Review · Reviewer_wGFt · 2024-07-13

**Soundness:** 4
**Presentation:** 4
**Contribution:** 3
**Rating:** 7
**Confidence:** 4

**Summary:**

This study proposes a fragment retrieval-augmented generation framework for molecular designs based on language models. The arm and linker vocabulary are constructed by fragments that have top average contribution to the given property. The hard fragments and a pool of soft fragments are retrieved from the vocabulary whose embeddings are fused and used for the generation. The generated molecules and the vocabulary are iteratively refined and augmented to further enhance the performance. The authors report competitive performance on multiple properties and generative tasks, including both single and multiple objectives.

**Strengths:**

This work formulates a novel framework for molecule designs and serves as a useful platform for future explorations. Though most of the techniques used in this study are not novel, the authors demonstrate effective ways of combining them to improve the performance. Specifically, the use of soft fragment pools and the genetic refinement make a good balance between exploration and exploitation.

The authors provide comprehensive evaluation, comparison and ablation results, and also show competitive performance in multi-objective optimization, which is more relevant to real-world drug discovery applications.

**Weaknesses:**

The work is overall solid and I don't have major concerns except some minor questions.

**Questions:**

1\. As the vocabularies are defined on the target property, is the fragment injection module also trained for each property? Would a universal model also work for the scenario?

2\. Line 242: the multi-objective optimization is performed using a unified score which is the product of all objectives. This may cause some problems such as high scores in one objective overshadowing others. How would this compare with, for example, selecting fragments that have higher scores for all three objectives?

3\. Table 3: as shown in Table 1, Genetic GFN does rather well in other tasks, so it would be better to also compare with it here.

4\. Line 275 and Table 8:  the no-GA setting has substantially higher diversity than all other settings. This is somewhat counterintuitive as GA expands the fragment vocabulary and could potentially lead to more diverse and novel molecules. Thus, the no-GA setting should have lower diversity just like it has lower novelty. What could be the cause of this observation?

5\. Fig 5c: what are the actual values of the results?

**Limitations:**

The authors have properly addressed the limitations.

---

> ### Author Rebuttal · Authors · 2024-08-06
>
> We sincerely appreciate your positive comments that our paper demonstrates an effective strategy for balancing exploration and exploitation, provides a comprehensive evaluation in many drug discovery tasks simulating real-world scenarios, and is overall solid. We address your questions below.
>
> ---
>
> **Comment 1**
>
> As the vocabularies are defined on the target property, is the fragment injection module also trained for each property? Would a universal model also work for the scenario?
>
> **Response 1**
>
> No, the training of the fragment injection module is **target property-agnostic**. As described in Section 3.2, we proposed to train the fragment injection module using a self-supervised objective that predicts the soft fragment that is most similar to the input fragment. The fragments used in training are independent of the target property, while the fragment vocabulary used in generation is constructed using the scoring function of Eq. (1). The purpose of the training fragment injection module is not to solve a particular drug discovery task, but to teach the whole model how to retrieve and fuse useful information to guide its generation. We will clarify this in the revised paper.
>
> ---
>
> **Comment 2**
>
> The multi-objective optimization is performed using a unified score which is the product of all objectives (line 242). How would this compare with selecting fragments that have higher scores for all three objectives?
>
> **Response 2**
>
> This is a good point! First of all, we used the unified score (Eq. (4)) to strictly follow the setting in previous works [1,2] for a fair evaluation. Second, in the experiment, we found that the two settings, (1) using the unified score and (2) first filtering out fragments that do not meet the QED and SA constraints and scoring them, yielded similar fragment vocabularies (out of top-50 fragments, 49, 50, 48, 50, and 50 were identical for the parp1, fa7, 5ht1b, braf, and jak2 tasks, respectively). Therefore, we believe that there is little difference between the two scoring schemes.
>
> ---
>
> **Comment 3**
>
> It would be better to compare with Genetic GFN in Table 3.
>
> **Response 3**
>
> We additionally compare our $f$-RAG with Genetic GFN [3] in the below table. $f$-RAG outperforms Genetic GFN by a very large margin, demonstrating its superiority as a universal method applicable to various drug discovery tasks. We appreciate your suggestion and will include this result in the revised paper.
>
> **Table: Novel top 5% docking score (kcal/mol) results.** The results are the means and standard deviations of 3 independent runs. Lower is better.
> | Method | parp1 | fa7 | 5ht1b | braf | jak2 |
> | --- | --- | --- | --- | --- | --- |
> | Genetic GFN | -9.227 ± 0.644 | -7.288 ± 0.433 | -8.973 ± 0.804 | -8.719 ± 0.190 | -8.539 ± 0.592 |
> | $f$-RAG (ours) | **-12.945** ± 0.053 | **-9.899** ± 0.205 | **-12.670** ± 0.144 | **-12.390** ± 0.046 | **-11.842** ± 0.316 |
>
> ---
>
> **Comment 4**
>
> The no-GA setting has higher diversity (Table 9).
>
> **Response 4**
>
> We note that optimization performance and molecular diversity are conflicting factors, as methods that generate non-optimized molecules can naturally easily have high diversity [3,4] (as described in Line 284). As shown in Table 8, $f$-RAG without GA showed AUC top-10 sum of 14.048, significantly worse than $f$-RAG’s 16.928. Therefore, the high diversity with low optimization performance of $f$-RAG without GA does not mean that the model can generate high-quality, diverse molecules, but rather that **the optimization is poor**. As you mentioned, GA helps $f$-RAG explore the chemical space beyond the initial fragment vocabulary, leading $f$-RAG to find better chemical optima and generate optimized molecules, whereas $f$-RAG without GA performs worse at finding chemical optima and generates diverse but low-quality molecules.
>
> ---
>
> **Comment 5**
>
> What are the actual values of the results in Fig 5c?
>
> **Response 5**
>
> We report the actual values of Figure 5(c) here. We will include the below table in the revised paper.
>
> **Table: PMO AUC top-10, top-100 diversity, top-100 novelty, and top-100 SA score results with different values of $\delta$ of the similarity-based fragment filter.**
> | Metric | $f$-RAG | $f$-RAG ($\delta$=0.8) | $f$-RAG ($\delta$=0.6) | $f$-RAG ($\delta$=0.4) |
> | --- | --- | --- | --- | --- |
> | Sum AUC | 16.928 | 16.648 | 16.262 | 15.765 |
> | Average diversity | 0.532 | 0.606 | 0.681 | 0.724 |
> | Average novelty | 0.800 | 0.778 | 0.751 | 0.796 |
> | Average SA score | 2.026 | 3.836 | 3.825 | 3.852 |
>
> ---
>
> **References**
>
> [1] Lee et al., Exploring chemical space with score-based out-of- distribution generation, ICML, 2023.
>
> [2] Lee et al., Drug discovery with dynamic goal-aware fragments, ICML, 2024.
>
> [3] Kim et al., Genetic-guided gflownets: Advancing in practical molecular optimization benchmark, arXiv, 2024.
>
> [4] Gao et al., Sample efficiency matters: a benchmark for practical molecular optimization, NeurIPS Datasets and Benchmarks, 2022.

---

> ### Comment · Reviewer_wGFt · 2024-08-12
>
> I appreciate the authors for the detailed response and updated results.

---

### Official Review · Reviewer_eome · 2024-07-13

**Soundness:** 2
**Presentation:** 3
**Contribution:** 2
**Rating:** 4
**Confidence:** 3

**Summary:**

Fragment-based drug discovery methods are limited in their exploration beyond existing database fragments, as they primarily reassemble or slightly modify the given fragments. This paper introduces a new approach, fragment retrieval-augmented generation (f-RAG), which retrieves two types of fragments—hard fragments and soft fragments—from a fragment vocabulary to achieve an improved exploration-exploitation to address this limitation.

**Strengths:**

This paper introduces a novel molecular generative framework that combines fragment-based drug discovery (FBDD) and retrieval-augmented generation (RAG). This paper proposes a retrieval augmentation strategy that operates at the fragment level, utilizing two types of fragments to provide fine-grained guidance. This approach aims to achieve a better exploration-exploitation trade-off and generate high-quality drug candidates.

**Weaknesses:**

1. This paper claims the f-RAG approach improve the exploration-exploitation trade-off. However, there is no experiment demonstrate this point.
2. No limitation is discussed.
3. Novelty and contribution is a concern, from the RAG part, it seems the main difference compared to Want et al. [42] is f-RAG dealing with fragment instead of molecule.
4. In addition, the critical part, SAFE-GPT[34], is a previous work.
5. f-RAG is built on a pre-trained backbone molecular language model, and it relies heavily on the generation performance of this backbone. This also means that the method delegates the challenging task of molecule generation to a large model.

**Questions:**

see weakness.

**Limitations:**

f-RAG is built on a pre-trained backbone molecular language model, and it relies heavily on the generation performance of this backbone. This also means that the method delegates the challenging task of molecule generation to a large model.

---

> ### Author Rebuttal · Authors · 2024-08-06
>
> We sincerely thank you for your comments. We appreciate your positive comments that our paper introduces a novel molecular generative framework that combines fragment-based drug discovery (FBDD) and retrieval-augmented generation (RAG). We address your concerns and questions below.
>
> ---
>
> **Comment 1**
>
> This paper claims the f-RAG approach improves the exploration-exploitation trade-off, but there is no experiment demonstrating this point.
>
> **Response 1**
>
> **The balance between exploiting chemical knowledge and exploring in the chemical space is essential to achieve good performance in molecular optimization problems**, and we have demonstrated the superiority of our proposed $f$-RAG through extensive molecular optimization experiments (Table 1 and Table 3). Furthermore, in addition to optimization performance, we also evaluated diversity, novelty, and synthesizability, other essential considerations in drug discovery. Here, diversity and novelty measure the ability of exploring the chemical space, and synthesizability measures the ability of exploiting chemical knowledge. As shown in Figure 1 and Table 2, $f$-RAG achieves the best optimization performance, diversity, and synthesizability, and the second best novelty. Overall, we can conclude that **$f$-RAG exhibits the best balance between exploration and exploitation across these essential considerations, demonstrating its applicability as a promising tool for drug discovery**.
>
> ---
>
> **Comment 2**
>
> No limitation is discussed.
>
> **Response 2**
>
> We discussed the limitations in Section A of the appendix.
>
> ---
>
> **Comment 3**
>
> Novelty is a concern. In the RAG part, the main difference from Wang et al. [1] is that f-RAG deals with fragments instead of molecules.
>
> **Response 3**
>
> Compared to RetMol, our proposed $f$-RAG is critically different in three aspects: (1) **$f$-RAG retrieves fragments instead of molecules, enabling much more fine-grained generative guidance**. There is a strong correlation between molecular structures and their activity, referred to as structure-activity relationship (SAR) [2], which means that there are important fragments in a given molecule that critically contribute to the target chemical property. Therefore, utilizing fragments instead of whole molecules results in more effective and chemically intuitive guidance. (2) **$f$-RAG utilizes two types of retrieval**, i.e., hard and soft fragment retrieval, while RetMol only performs soft retrieval (of molecules). In this way, $f$-RAG can effectively balance between exploitation of current chemical knowledge and exploration in the chemical space. (3) **$f$-RAG generates molecules in a one-shot manner, while RetMol relies on iterative refinement** that uses retrieved guidance to refine noise over multiple iterations (80 iterations in the paper). This is a significant drawback for many drug discovery problems where oracle calls are expensive and oracle budgets must be considered. Due to these differences, $f$-RAG outperforms RetMol by a very large margin (Table 3).
>
> ---
>
> **Comment 4**
>
> f-RAG is built on a pre-trained backbone molecular language model, SAFE-GPT [3], and it relies heavily on the generation performance of this backbone. This also means that the method delegates the challenging task of molecule generation to a large model.
>
> **Response 4**
>
> As we have mentioned in Limitations (Section A), our proposed $f$-RAG is built on a pre-trained backbone molecular language model, SAFE-GPT. This design choice that utilizes a pre-trained LLM is a very popular strategy across many domains [1,3]. **This strategy is a large advantage rather than a disadvantage, as it lets the lightweight fragment injection module take care of the relatively easy task of fragment retrieval augmentation**. As shown through the extensive experiments in Section 4, this strategy makes $f$-RAG **a simple but powerful method** to solve various drug discovery tasks. Furthermore, it also enables **very efficient and fast training** of $f$-RAG. As described in Section D.1, the fragment injection module, the only part of $f$-RAG that requires training, is very lightweight. The module has 2,362,368 trainable parameters, which correspond to only 2.64% of the total parameters of 89,648,640. As described in Section D.4, this allows us to train $f$-RAG in less than 4 hours using a single GeForce RTX 3090 GPU, while training of SAFE-GPT takes 7 days using 4 NVIDIA A100 GPUs [4]. Moreover, we emphasize that **the high performance of $f$-RAG cannot be achieved by the backbone large model alone**. For example, $f$-RAG without our proposed fragment retrieval showed AUC top-10 sum of 15.395 while the full $f$-RAG showed 16.928 (Table 8). Overall, we believe that combining the generative power of a large pre-trained model and a novel fragment retrieval augmentation strategy is an important contribution that can have practical impact.
>
> **We appreciate your detailed feedback. We hope our response addresses your concerns and that you consider upgrading your rating. We are happy to elaborate further if there are any remaining concerns.**
>
> ---
>
> **References**
>
> [1] Wang et al., Retrieval-based controllable molecule generation, ICLR, 2023.
>
> [2] Crum-Brown et al., The connection of chemical constitution and physiological action. Trans R Soc Edinb, 25(1968-1969):257, 1865.
>
> [3] Yu et al., Enzyme function prediction using contrastive learning, Science 379.6639: 1358-1363, 2023.
>
> [4] Noutahi et al., Gotta be safe: a new framework for molecular design, Digital Discovery, 3(4):796–804, 2024.

---

> > ### Comment · Reviewer_eome · 2024-08-13
> >
> > Thank you for your response. My concerns regarding the contribution and novelty of this paper still remain, so I will keep my score unchanged.

---

### Author Rebuttal · Authors · 2024-08-06

Dear reviewers, we sincerely appreciate your constructive comments. There were a number of comments that will help us strengthen our paper, and we will be sure to incorporate them into the revision.

We have had a few questions about the difference of our proposed $f$-RAG compared to RetMol [1], so we would like to clarify this in the global response. Compared to RetMol, our proposed $f$-RAG is critically different in three aspects: (1) **$f$-RAG retrieves fragments instead of molecules, enabling much more fine-grained generative guidance**. There is a strong correlation between molecular structures and their activity, referred to as the structure-activity relationship (SAR) [2], which means that fragments are building blocks of molecules that critically contribute to their target chemical property. Therefore, utilizing fragments instead of whole molecules enables compositionally in generation and results in more effective and chemically intuitive guidance. (2) **$f$-RAG utilizes two types of retrieval**, i.e., hard and soft fragment retrieval, while RetMol only performs soft retrieval (of molecules). In this way, $f$-RAG can effectively balance between exploitation of current chemical knowledge and exploration in the chemical space. (3) **$f$-RAG generates molecules in a one-shot manner, while RetMol relies on iterative refinement** that uses retrieved guidance to refine noise over multiple iterations (80 iterations in the paper). This is a significant drawback for many drug discovery problems where oracle calls are expensive and oracle budgets must be considered. Due to these differences, $f$-RAG outperforms RetMol by a very large margin (Table 3).

---

**References**

[1] Wang et al., Retrieval-based controllable molecule generation, ICLR, 2023.

[2] Crum-Brown et al., The connection of chemical constitution and physiological action. Trans R Soc Edinb, 25(1968-1969):257, 1865.

---

### Decision · Program_Chairs · 2024-09-25

**Decision:**

Accept (poster)

**Comment:**

See comment to SAC